# Robust activity-dependent mitochondrial calcium dynamics at the AIS is dispensable for action potential generation

Koen Kole[1,3] 🆔 and Maarten H.P. Kole[1,2] 🆔

[1]*Department of Axonal Signaling, Netherlands Institute for Neuroscience, Royal Netherlands Academy of Arts and Science, Amsterdam, The Netherlands*
[2]*Cell Biology, Neurobiology and Biophysics, Department of Biology, Faculty of Science, University of Utrecht, Utrecht, The Netherlands*
[3]*Donders Center for Neuroscience, Donders Institute for Brain, Cognition and Behaviour, Nijmegen, The Netherlands*

Handling Editors: Nathan Schoppa & Samuel Young

The peer review history is available in the Supporting information section of this article (https://doi.org/10.1113/JP289290#support-information-section).

**Abstract figure legend** In cortical layer (L)5 pyramidal neurons we found that mitochondria in the axon initial segment (AIS) are distributed non-uniformly and display robust calcium ($Ca^{2+}$) uptake during action potential (AP) firing. Because the AIS is the site for the initiation of APs, which are shaped by cytosolic $Ca^{2+}$ levels, we hypothesized that mitochondrial $Ca^{2+}$ buffering at the AIS could impact the AP waveform. However, when we pharmacologically blocked mitochondrial $Ca^{2+}$ uptake, we only detected an increase in the duration of the afterhyperpolarization (AHP), while the AP initiation and waveforms were unaltered. These data suggest that mitochondria at the AIS play other, non-electrical, roles.

**Abstract** Mitochondria are diverse and multifaceted intracellular organelles regulating oxidative energy supply, lipid metabolism and calcium ($Ca^{2+}$) signalling. In neurons the spatial sequestration of cytoplasmic $Ca^{2+}$ by mitochondria plays a critical role in determining activity-dependent spine plasticity, shaping the presynaptic transmitter release characteristics and contributing to sustained action potential firing. Here, we tested the hypothesis that mitochondria at the axon initial segment (AIS) affect the microdomain cytoplasmic $Ca^{2+}$ transients, thereby regulating $Ca^{2+}$-dependent voltage-gated ion channels at the plasma membrane and initiation of action potentials. Using 3D electron microscopy reconstructions and virally injecting genetically encoded fluorescence indicators we visualized the ultrastructure and distribution of mitochondria selectively in thick-tufted layer 5 pyramidal neurons. We found that most mitochondria were stably clustered to the proximal AIS, while few were observed at distal sites. Simultaneous two-photon imaging of action potential-dependent cytoplasmic and mitochondrial $Ca^{2+}$, combined with electrophysiological recordings showed that AIS mitochondria exhibit powerful activity-dependent cytosolic $Ca^{2+}$ uptake. However, while intracellular application of the mitochondrial $Ca^{2+}$ uniporter inhibitor Ru360 fully blocked mitochondrial $Ca^{2+}$ import and increased the slow afterhyperpolarization duration, it did not affect action potential input–output function, action potential dynamics nor the ability to produce high-frequency burst output. Together, the results indicate that AIS mitochondria are dispensable for temporal and rate encoding, suggesting that mt-$Ca^{2+}$ buffering at the AIS may be involved in non-electrical roles.

(Received 24 May 2025; accepted after revision 11 February 2026; first published online 9 March 2026)

**Corresponding author** M. H.P. Kole: Department of Axonal Signaling, Netherlands Institute for Neuroscience, Royal Netherlands Academy of Arts and Science, Meibergdreef 47, 1105 BA, Amsterdam, The Netherlands. Email: m.kole@nin.knaw.nl

### Key points

- Mitochondrial $Ca^{2+}$ buffering controls multiple $Ca^{2+}$-dependent intracellular processes and their subcellular location of the organelles defines local physiological properties in neurons.
- Recent studies implicate mitochondrial $Ca^{2+}$ uptake in the slow afterhyperpolarization and maintenance of action potential firing.
- Using electron microscopy and virally delivered genetically encoded tools we examined mitochondria in the layer 5 pyramidal neuron axon initial segment (AIS), the site where action potentials initiate, and found that cytoplasmic $Ca^{2+}$ influx is powerfully buffered by proximally clustered mitochondria.
- Electrophysiological recordings during the block of the mitochondrial calcium uniporter reveal a role in the slow afterhyperpolarization, while AIS action potential initiation and action potential waveforms are independent from mitochondria.
- These findings indicate AIS mitochondria under physiological conditions exert non-electrical roles.

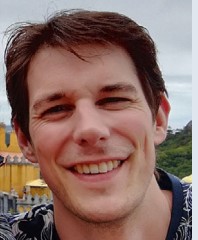

Koen Kole obtained his PhD on cellular and molecular neuroscience at the Radboud University in Nijmegen, the Netherlands. He then went on to perform his postdoctoral studies at the Netherlands Institute for Neuroscience in Amsterdam. Currently he works as a postdoc at the Donders Institute in Nijmegen. Koen uses a combination of molecular, imaging and electrophysiological techniques to study the inter- and subcellular functional heterogeneity of neuronal organelles, and how they support neuronal function.

## Introduction

Mitochondria are organelles of the eukaryotic cell fundamentally tasked with the production of the cellular energy carrier adenosine triphosphate (ATP). Emerging insights have shown additional and versatile functions of mitochondria including lipid synthesis, regulation of cell death and, notably, calcium ($Ca^{2+}$) homeostasis (Rizzuto et al., 2012). The multifaceted functionality of mitochondria is particularly clear in neurons, which are highly dependent on both ATP and $Ca^{2+}$ homeostasis to enable the faithful propagation of action potentials (APs) and synaptic vesicle release (Attwell & Laughlin, 2001; Burgoyne, 2007; Hallermann et al., 2012). In neurons, the strong morphological polarization and compartmentalization of the cytoarchitecture is also reflected by the diversity in the mitochondrial pool. For instance, mitochondria in the dendrites are elongated and support spine plasticity (Lewis et al., 2018; Rangaraju et al., 2019), whereas axonal mitochondria are short and sustain synaptic vesicle release (Ashrafi et al., 2017, 2020). Beyond ATP synthesis, the mitochondrial $Ca^{2+}$ (mt-$Ca^{2+}$) buffering within axons plays an important role in local $Ca^{2+}$-dependent processes. Within presynaptic terminals, mitochondria take up $Ca^{2+}$ and sequester it, lowering local cytosolic $Ca^{2+}$ levels and reducing synaptic vesicle release rates (Kwon et al., 2016; Lewis et al., 2018). Mt-$Ca^{2+}$ buffering is thus an effective means by which neurons can regulate their cytosolic $Ca^{2+}$ levels within subcompartments, allowing high spatial and temporal control over $Ca^{2+}$-dependent pathways.

Interestingly, by $Ca^{2+}$ buffering, mitochondria also impact the electrophysiological membrane properties of neurons. Cytosolic $Ca^{2+}$ entering the mitochondrial matrix drives ATP production via oxidative phosphorylation (Rizzuto et al., 2012; Szibor et al., 2020), thereby supplying energy to, for example, the sodium ($Na^+$)/potassium ($K^+$) ATPase (Stoler et al., 2022; Zampese et al., 2022). Such fuelling of the $Na^+$/$K^+$ ATPase restores ionic concentrations and sustains spiking rates (Lin et al., 2019; Stoler et al., 2022; Zampese et al., 2022). Previous studies in cortical pyramidal neurons or vasopressin neurons of the hypothalamus show that mitochondria-dependent regulation of the membrane excitability adjusts the firing rate as well as the $Ca^{2+}$-dependent slow afterhyperpolarization (AHP) current (Groten & MacVicar, 2022; Kirchner et al., 2024; Styr et al., 2019). It remains to be tested whether $Ca^{2+}$ buffering by mitochondria also impacts other electrophysiological properties, such as firing frequency, bursting or AP waveform. Moreover, it is unknown whether the previous observations result from the collective $Ca^{2+}$ buffering capability of all mitochondria in a given neuron, or if there is a specific locus where mt-$Ca^{2+}$ buffering controls neuronal electrophysiological properties. One

highly specialized subcompartment of the neuron is the axon initial segment (AIS), where APs are initiated by the clustering of high densities of voltage-dependent ion channels (Jenkins & Bender, 2024; Kole & Stuart, 2012). While the membrane excitability at the AIS axolemma is dominated by the voltage-dependent $Na^+$ and $K^+$ channels, AP generation is also shaped by voltage-dependent $Ca^{2+}$ influx and cytoplasmic $Ca^{2+}$ concentrations. For example, blocking voltage-gated $Ca^{2+}$ channels locally at the AIS increases the AP threshold and delays AP initiation within high-frequency bursts (Bender & Trussell, 2009). Furthermore, $Ca^{2+}$ not only charges the membrane but also acts as a downstream signalling molecule: axonal $Ca^{2+}$-activated $K^+$ ($K_{Ca}$) channels regulate AP decay time and decrease neuronal excitability (Filipis et al., 2023; Gründemann & Clark, 2015; Yu et al., 2010). Several voltage-gated ion channels possess $Ca^{2+}$-calmodulin sensing domains, mediating $Ca^{2+}$-dependent regulation of kinetics of voltage-gated sodium channels (Ben-Johny et al., 2014; Wang et al., 2014) or closure of the Kv7 channel (Bernardo-Seisdedos et al., 2018; Delmas & Brown, 2005; Martinello et al., 2015). The $Ca^{2+}$-dependent regulation of ion channels requires neurons to carefully control their cytoplasmic $Ca^{2+}$ levels at the AIS, which can be achieved in part by sequestering it internally and regulating its release via stores (Verkhratsky & Petersen, 1998).

Mitochondria have been reported in the AIS, but their function in this domain remains poorly understood (Dimova & Markov, 1976; Kole et al., 2022; Li et al., 2004; Tjiang & Zempel, 2022). The main entry route for $Ca^{2+}$ into the mitochondrial matrix is via the mitochondrial calcium uniporter (MCU), a selective $Ca^{2+}$ channel residing in the mitochondria inner membrane. The MCU has a low $Ca^{2+}$ affinity, activated with cytoplasmic $Ca^{2+}$ concentrations as low as ~1 μM (Payne et al., 2017). AIS mitochondria in neocortical layer 5 (L5) pyramidal neurons may rapidly encounter such cytosolic $Ca^{2+}$ levels since even a single AP raises cytoplasmic $Ca^{2+}$ by ~1 μM (Hanemaaijer et al., 2020). Here, we tested the hypothesis that mitochondrial $Ca^{2+}$ buffering at the AIS regulates AP initiation and generation. We examined the anatomical distribution of L5 pyramidal neuron mitochondria by using 3D electron microscopy (EM) reconstructions and functionally studied their role by using genetically encoded tools to visualize mitochondria and their $Ca^{2+}$ uptake, combined with electrophysiological recordings and pharmacological manipulation. We found that the AIS contains a gradient of mitochondria which are stably located and exhibit strong $Ca^{2+}$ buffering. Intracellular application of the MCU pore complex inhibitor Ru360, however, did not alter AP firing properties. Our results reveal a striking axonal compartmentalization of mitochondrial distribution and $Ca^{2+}$ buffering and

suggest that mt-Ca$^{2+}$ buffering at the AIS is dispensable for AP generation in L5 pyramidal neurons.

## Materials and methods

### Ethical approval

All animal procedures were performed after evaluation by the Royal Netherlands Academy of Arts and Sciences (KNAW) Animal Ethics Committee (DEC) and Centrale Commissie Dierproeven (CCD, licence AVD8010020172426). The specific experimental designs with animals were evaluated and monitored by the Animal Welfare Body (IvD, protocols NIN19.21.09, NIN19.21.12 and NIN21.2108).

### Mice

The mouse strain used for this research project was B6.FVB(Cg)-Tg(Rbp4-cre)KL100Gsat/Mmucd, RRID:MMRRC_03 7128-UCD, obtained from the Mutant Mouse Resource and Research Centre (MMRRC) at the University of California at Davis, an NIH-funded strain repository. This strain, here called *Rbp4-Cre*, was made from the original strain MMRRC:03 2115 donated by Nathaniel Heintz, Ph.D., The Rockefeller University, GENSAT and Charles Gerfen, Ph.D., National Institutes of Health, National Institute of Mental Health. For a subset of electrophysiological recordings, we used wild-type mice (C57BL/6JRj; RRID:IMSR_RJ:C57BL-6JRJ). Male or female mice (8–11 weeks old) were kept at a 12 h day–night cycle with *ad libitum* access to food pellets and water. Cages were open or individually ventilated cages with corncob bedding. Mice were housed together with at least one cage mate. Ambient temperature was maintained at 20–24°C, humidity at 45–65%.

### Adeno-associated virus production

Plasmids to generate mt-GFP or mt-GCaMP6f adeno-associated virus (AAV) were generated as described previously (Kole et al., 2022). For AAV production, HEK293-T cells of low (<25) passage were maintained in Dulbecco's modified Eagle's medium (DMEM, Thermo Fisher Scientific #31 966-047). The medium contained 10% fetal calf serum (FCS, Thermo Fisher Scientific A4766801) and 5% penicillin–streptomycin (Pen-Strep, Thermo Fisher Scientific #15 140 122), cells were maintained at 37°C and 5% CO$_2$. For virus production, cells were seeded in 15 cm dishes at a density of $1$–$1.25 \times 10^7$ cells per dish. The following day, 1–2 h before transfection, the DMEM was replaced by fresh Iscove's modified Dulbecco's medium (IMDM, Sigma-Aldrich I-3390) containing 10% FCS, 5% Pen-Strep and 5% glutamine (Thermo Fisher Scientific #25 030 081). For transfection, AAV rep/cap, AAV helper (mt-GFP: AAV5; mt-GCaMP6f: AAV1) and transfer plasmids were mixed and diluted in saline before mixing with saline-diluted polyethylenimine (Polysciences #23 966-2) and brief vortexing. After incubation (20–25 min), the transfection mix was added to the culture plates in a drop-wise fashion. After 16 h, the medium was refreshed, after which the cells were left for an additional 72 h. Then, medium was discarded and cells were then collected. Lysis to release AAVs was achieved via three freeze–thaw cycles. Cell lysate was then loaded on an iodixanol gradient (60%, 40%, 25% and 15% iodixanol, ELITechGroup #1 114 542) in Beckman Quick-Seal Polyallomer tubes (Beckman-Coulter #342 414). Centrifugation was performed at 16°C and 69,000 rpm (488,727.6 *g*) for 1 h and 10 min in a Beckman-Coulter Optima XE-90 Ultracentrifuge using a Type 70 Ti rotor. The virus-containing fraction was then extracted from the tubes and AAVs were then concentrated in Dulbecco's Phosphate Buffered Saline (D-PBS) + 5% sucrose using Amicon Ultra-15 (100 K) filter units (Merck Millipore UFC910024) at 3220 *g*. To ensure complete replacement of iodixanol with D-PBS + 5% sucrose, at least four rounds of centrifugation were used. Viral titres were determined using quantitative PCR (titres: mt-GFP-DIO, $4.43 \times 10^{13}$ gc/mL; mt-GCaMP6f-DIO, $3.44 \times 10^{13}$ gc/mL). Viral aliquots were stored at −80°C until further use.

qPCR primers, recognize AAV2 inverted terminal repeats (ITRs): 5′-GGAACCCCTAGTGATGGAGTT-3′ 5′-CGGCCTCAGTGAGCGA-3′

### Viral injection

Viral injections were typically performed at 8–9 weeks of age. Mice were anaesthetised using isoflurane (induction, 3%; maintenance, 1.2–1.5%) after which they received 5 mg/kg Metacam subcutaneously. Using a heating pad, the body temperature was monitored and maintained at 37°C. To prevent the eyes from drying out, eye ointment was applied. Then, the head was shaved using electronic clippers and hair removal cream, after which it was placed in a stereotaxic frame (Kopf). An incision was made in the skin along the midline. Before removing the periost, lidocaine (10%) was administered locally. Small (<1 mm) bilateral craniotomies were made at −0.5 mm caudally from Bregma and 2.5 mm laterally from the midline. Care was taken not to damage the dura mater. Using a Nanoject III (Drummond) fitted with a sharp glass pipette, 40–50 nL of virus was injected at 1 nL/s and at a depth of 450 μm. Approximately 3 min after finishing the injection, the needle was retracted slowly. Bone wax was applied to the craniotomies and the skin was carefully sutured

before mice were allowed to recover. During the 3–5 days following surgery, mice were monitored closely. Their weight, locomotion and overall wellbeing were checked.

## Acute slice preparation

After 2–3 weeks of virus expression, mice were deeply anaesthetized using pentobarbital (50 mg/kg, intraperitoneal injection) and transcranially perfused with ice-cold, carbogenated (95% $O_2$, 5% $CO_2$) cutting artificial cerebrospinal fluid (cACSF; 125 mM NaCl, 3 mM KCl, 6 mM $MgCl_2$, 1 mM $CaCl_2$, 25 mM glucose, 1.25 mM $NaH_2PO_4$, 1 mM kynurenic acid and 25 mM $NaHCO_3$). The brain was quickly dissected out, after which 400 μm thick parasagittal slices were cut using a Vibratome (1200S, Leica Microsystems) all the while keeping the brain submerged in ice-cold carbogenated cACSF. Slices were immediately transferred to a holding chamber containing carbogenated cACSF at 35°C where they were kept for 35 min to recover. After this period, they were allowed to return to room temperature for at least 30 min before starting experiments.

## Electrophysiology and two-photon imaging

Slices were transferred to a recording chamber with continuous perfusion (1–2 mL per min) of carbogenated recording ACSF (rACSF, 125 mM NaCl, 3 mM KCl, 1 mM $MgCl_2$, 2 mM $CaCl_2$, 10 mM glucose, 5 mM L-Lactate, 1.25 mM $NaH_2PO_4$ and 25 mM $NaHCO_3$). Bath temperature was maintained at 32°C. Glass pipettes with an open tip impedance of 6–7 MΩ were filled with an intracellular solution containing (130 mM K-Gluconate, 10 mM KCl, 10 mM HEPES, 4 mM Mg-ATP, 0.3 mM $Na_2$-GTP, 10 mM $Na_2$-phosphocreatine; pH ∼7.25, osmolality ∼280 mOsmol/kg). For mt-$Ca^{2+}$ inhibition experiments, intracellular solution was supplemented with Ru360 (20 μM; Sigma-Aldrich #557 440, 20 mM stock in water and stored at −20°C) or equal volume of water (control). In a subset of experiments, the intracellular solution was supplemented with 5 mg/mL biocytin (Sigma-Aldrich, B4261) and 50 μM Atto-594 (Sigma-Aldrich, A08637). Whole-cell recordings were made using a patch-clamp amplifier (Multiclamp 700B, Axon Instruments, Molecular Devices, RRID: SCR_01 8455) operated by AxoGraph X software (version 1.5.4; RRID: SCR_01 4284). AP trains were evoked using 700 ms step pulses, from −250 pA to +1000 nA with increments of 50 pA. Starting below spike threshold, 3 ms incremental (2.5–5 pA) step pulses were used to evoke single APs. Voltage was digitally sampled at 100 Hz using an AD/DA converter (ITC-18, HEKA Elektronik GmbH). During current-clamp experiments, the access resistance (range: 10–15 MΩ) was fully compensated using bridge balance and capacitance neutralization of the amplifier and monitored carefully during recordings. In cases were a maximum of 20 MΩ was exceeded, cells were excluded from the analysis. Electrophysiological data were excluded if cells had an unstable resting membrane potential. Membrane potentials in this study are corrected for a −14 mV junction potential of the intracellular recording solution. Somatic single-cell recordings were made from mt-GFP$^+$ or mt-GCaMP6$^+$ cells, which were visualized using a two-photon (2P) laser-scanning microscope (Femto3D-RC, Femtonics Inc., Budapest, Hungary). Imaging was controlled using MES software (Femtonics Inc., Budapest, Hungary, version 6.3.7902). To visualize mt-GFP/mt-GCaMP6f, a Ti:Sapphire pulsed laser (Chameleon Ultra II, Coherent Inc., Santa Clara, CA, USA) was tuned to 770 nm for 2P excitation. Fluorescent signals were detected using two photomultipliers (PMTs, Hamamatsu Photonics Co., Hamamatsu, Japan), one for mt-GFP and for Cal590. For motility imaging, healthy appearing somata and corresponding AISs were identified using brightfield light as well as 2P mt-GFP imaging, and z-stacks were acquired every 10 s for 8–12 min. The Image Stabiliser plugin for FIJI was used for drift correction. For $Ca^{2+}$ imaging, mt-GCaMP6f$^+$ cells were targeted for single-cell patch clamping and filled with Cal-590 (20 μM; Sigma 0 8637). Next, z-stacks were made with the laser tuned to 800 nm to identify the axon and its subcompartments (AIS, myelinated internode and nodes). Mt- and cyt-$Ca^{2+}$ responses were then simultaneously visualized with the laser tuned to 940 nm and recorded at ∼20 Hz imaging frequency. Optical and electrophysiological recordings were synchronized via a TTL pulse from the microscope to the amplifier. $Ca^{2+}$ responses were analysed using a custom-written Matlab script (Kole et al., 2022). Briefly, regions of interest were selected, from which background was subtracted, bleach correction and smoothing were applied, and ΔF/F was calculated.

## Immunohistochemistry

For staining of biocytin-filled cells, upon completion of experiments acute slices were immediately placed into 4% PFA in PBS for 20–25 min, followed by three washes of 10 min with PBS. Next, blocking was performed in PBS containing 1–2% Triton-X and 10% normal goat serum for 2 h. After blocking, sections were moved to blocking buffer containing primary antibodies (see Table A1) and were incubated overnight at room temperature. Following three 10 min washes in PBS, sections were transferred to PBS containing secondary antibodies and were incubated for 2 h at room temperature or overnight at 4°C. The tissue was finally washed again in PBS three times for 10 min before mounting using

FluorSave mounting medium (Merck Millipore #345 789). For immunostainings against myelin basic protein, the staining protocol was adjusted: blocking was done for 1 h at 37°C and 1 h at room temperature and using 1% Triton, tissue was washed only once in PBS per washing step, and secondary antibodies were incubated for 1 h at 37°C and 1 h at room temperature. For immunostainings on sections without biocytin-filled cells, 400 μm PFA-fixed sections were first cryoprotected by placing them into 30% sucrose–PBS solution until fully saturated. A sliding freezing microtome (Zeiss Hyrax S30; temperature controlled by a Slee Medical GmbH MTR fast cooling unit) was used to cut 40 μm sections, which were either placed in PBS for immediate use or stored at −20°C in cryoprotectant solution (30% ethylene glycol, 20% glycerol, 0.05 M phosphate buffer) until further use. Immunostaining protocol was the same as for the 400 μm sections, but the blocking buffer contained 0.5% Triton-X and incubation duration for secondary antibodies was 2 h at room temperature. All steps are performed with gentle shaking, except for the 37°C incubation steps. For analysis of normal and demyelinated subcortical white matter, *Rbp4-Cre* mice injected with AAV1-EF1a-mCherry-DIO and AAV5-EF1a-mtGFP-DIO (mixed 1:1) were first perfused with 1× PBS followed by 4% PFA-PBS. Brains were then dissected out and allowed to fix O/N in 4% PFA-PBS after the brains were cryoprotected using 30% sucrose–PBS and processed into 40 μm coronal sections as described above.

### Confocal microscopy

For confocal imaging, a Leica SP8 X confocal laser-scanning microscope controlled by Leica Application Suite AF (version 3.5.7.23225) was used. Biocytin-filled cells were imaged with a 63× oil-immersion lens. The tile-scan function was used with automated sequential acquisition of multiple channels enabled, and step sizes in the *z*-axis were 0.3–0.75 μm, and images were collected at a 2048 × 2048 pixel resolution at 100–150 Hz. Axonal reconstructions and quantification of mitochondrial density were performed manually using Neurolucida Software (MBF Bioscience, version 2019.2.1 or 2020.1.3, 64 bit, RRID: SCR_0 01775). Care was taken to include only mitochondria of which the mt-GFP signal was located clearly inside the cytosol and followed the path of the reconstructed neurite. Upon completion of tracing, analysis of reconstructions was performed using Neurolucida Explorer (MBF Bioscience, 2019.2.1, RRID: SCR_01 7348). FIJI (FIJI 64 bit; ImageJ version 1.53q; RRID: SCR_0 02285) was used to extract partial images for use in figures and generation of the kymograph in Fig. 3. Motile mitochondria were identified by eye,

mitochondria were considered stable if they displaced less than 2 μm.

### Three-dimensional electron microscope data analysis

Three-dimensional EM data was obtained from the Microns dataset (JAX stock 02 3527 and 03 1562, respectively). L5 pyramidal neurons were identified based on their distance from the pia as well as their morphology (i.e. their pyramidal-shaped soma, large apical dendrite and axon projecting into the white matter). EM volumes containing the AIS and the first internode, as well as their cytosolic segmentation were then downloaded, after which the mitochondria were segmented and the segment length traced manually using the Volume Annotation and Segmentation Tool (VAST version 1.4.1) (Berger et al., 2018; MICrONS Consortium, 2025) (www.microns-explorer.org). This dataset consists of a 1 mm³ EM block of primary visual cortex of one P87 male mouse expressing GCaMP6s in excitatory neurons (*Slc17a7*-Cre and Ai162 heterozygous transgenic lines). VAST Tools Matlab scripts were used for length and volume measurements and exports. Exported 3D models were then rendered using 3ds Max (Autodesk, version 25.0.0.997, SCR_01 4251). See Table A2 for cells used in the analysis.

### Statistics and reproducibility

Prism (Graphpad, version 8.4.3, RRID: SCR_0 02798) was used for all statistical comparisons. Outliers were detected using the Robust regression and Outlier removal (ROUT) method and excluded from the analysis. Dataset normality was determined using the D'Agostino & Pearson or Shapiro-Wilk tests. If data deviated significantly from a normal distribution we used non-parametric tests. For comparisons between groups, we used a two-tailed unpaired *t* test (normal data) or a two-tailed Mann–Whitney test (non-normal data). To test interactions between groups and treatments two-way ANOVA was used; Bonferroni's *post hoc* test (normal data), Dunn's *post hoc* test (non-normal data) or Fisher's *post hoc* test was used for multiple comparisons. To avoid overpowering of non-nested statistical tests, a nested *t* test was used when large numbers of datapoints were involved (i.e. mitochondrial contours). Figure legends contain *P* values and *n* numbers and whether the latter signify mice, cells or mitochondria. Means are presented with SD. To ensure reproducibility, data were collected from multiple cells from multiple mice: experiments were replicated in at least four cells from three mice. The only exception is the 3D EM reconstructions which were done using data obtained from one mouse. Cells were randomly infused

with either control or Ru360-containing intracellular solution.

## Results

### Mitochondria densely populate the AIS

To examine the distribution of mitochondria in L5 pyramidal neuron AISs at ultrastructural detail we made use of a publicly available annotated 3D EM dataset of the adult mouse primary visual cortex (Consortium, 2025). We randomly selected thick-tufted L5 pyramidal neurons and segmented mitochondria at the AIS (excluding the first 3 μm representing the axon hillock, the transition from soma into the AIS) and the first myelinated internode, quantifying their occupancy, size and density (Fig. 1*A*–*C*). At the AIS, defined as the unmyelinated region between the axon hillock and first paranodal loops, mitochondria occupied mostly the proximal AIS and significantly reduced in probability at the distal 10 μm, in line with previous reports (Tamada et al., 2021; Tjiang & Zempel, 2022) (Fig. 1*Da*). Within internodes, mitochondria tended to avoid the first ∼2 μm representing the paranode but were otherwise uniformly distributed (Fig. 1*Db*). We observed that ∼75% of first nodes of Ranvier (branch points after the first internode) harboured mitochondria (Fig. 1*Db*). Considering the AIS as a whole, mitochondria were present at significantly higher densities compared with the first internode (average ± SD 0.77 ± 0.24 *vs.* 0.38 ± 0.10 mitochondria/μm; paired *t* test *P* = 0.0089) and were comparable in size (AIS, 0.11 ± 0.05; internode, 0.14 ± 0.06 μm³; paired *t* test *P* = 0.0822). Given the declining mitochondrial occupancy in the distal AIS, we quantified the density and size of mitochondria in the first and second 50% of the total AIS length (proximal and distal AIS, respectively). The data showed that mitochondria were significantly more abundant within the proximal AIS at over twice the density (Fig. 1*E*; average prox AIS, 1.14; dist AIS, 0.45; internode, 0.38 mitochondria/μm; repeated one-way ANOVA *P* = 0.0011). In addition, mitochondria within the distal AIS were significantly smaller than in the internode (Fig. 1*F*; prox AIS, 1.12 ± 0.02; dist AIS, 0.05 ± 0.02; internode, 0.15 ± 0.02 μm³; repeated one-way ANOVA *P* = 0.0132). This also tended to be true for the mitochondrial volume relative to that of the axon, but this did not reach statistical significance in *post hoc* tests (Fig. 1*G*; prox AIS, 0.1191 ± 0.0183; dist AIS, 0.0680 ± 0.0604; internode, 0.1261 ± 0.0249 v/v; repeated one-way ANOVA *P* = 0.0369). Together, these results indicate that mitochondria densely populate the proximal AIS.

To test further the distribution of mitochondria in molecularly identified AISs we next studied acute slices using previously developed Cre-dependent viral tools to fluorescently label mitochondria (Kole et al., 2022). *Rbp4*-Cre mice, which express Cre recombinase selectively in L5 pyramidal neurons (Gerfen et al., 2013), were injected in the primary somatosensory cortex with an AAV vector to express Cre-dependent mitochondria-targeted green fluorescent protein (mt-GFP). Immunofluorescence staining showed that this viral approach effectively and specifically labels mitochondria in L5 pyramidal neurons (Fig. 2*A*). We then applied immunolabelling of the AIS-associated cytoskeletal protein ßIV-spectrin, to identify mitochondria within the molecularly identified AIS (Fig. 2*A*–*C*). In acute brain slices, we made patch-clamp recordings from L5 pyramidal neurons to fill the cytoplasm with biocytin, allowing detailed reconstruction of mt-GFP expressing (mt-GFP⁺) mitochondria at the AIS and first internode. In accord with the 3D EM data, the AIS contained mitochondria at significantly higher density in the proximal AIS, and mitochondria numbers declining with distance from the soma (Fig. 2*C* and *D*; prox AIS, 0.45 ± 0.06; dist AIS, 0.23 ± 0.04; internode, 0.22 ± 0.01 mitochondria/μm; repeated one-way ANOVA *P* = 0.0033). Compared with 3D EM, our immunohistochemistry data indicated lower mitochondrial densities (on average 0.36 *vs.* 0.67 mitochondria/μm; unpaired *t* test, *P* = 0.0118). This is likely explained by the much higher spatial resolution that EM offers (4 nm), allowing the identification of smaller mitochondria that are below the detection threshold of confocal microscopy (>400 nm). In addition, we observed that mitochondria were often situated closely together, likely beyond the resolution limits of confocal or two-photon microscopy. It is therefore important to note that, in the next sections, the mitochondria we detected may in fact represent clusters rather than individual organelles.

We next asked whether the high mitochondrial density at the proximal AIS represented a stable population or if they were mitochondria merely transported via microtubules within the AIS antero- or retrogradely between the soma and the distal axon (Misgeld & Schwarz, 2017). We therefore made live 2P recordings of mitochondria at the AIS in acutely prepared brain slices of mitoGFP-expressing mice (Fig. 3*A*). Somata and AISs were readily recognizable by their high mitochondrial content, negating the necessity of a cytosolic marker to identify them. Although some mitochondria were motile, we found that the majority (73.64 ± 21.43%) was stably present at the AIS for the duration of the imaging session (8 to 10 min, Fig. 3*B*), in agreement with previous reports (Tjiang & Zempel, 2022). Motile mitochondria predominantly moved in an anterograde direction (Fig. 3*C*; 89.35 ± 18.53%) and were significantly smaller than stable mitochondria (Fig. 3*D*; nested *t* test, *P* < 0.0001; 0.36 ± 0.21 *vs.* 1.46 ± 0.89 μm²). We found no correlation between size and speed (Fig. 3*E*; Pearson *r* = −0.08265, *P* = 0.6881). Taken together,

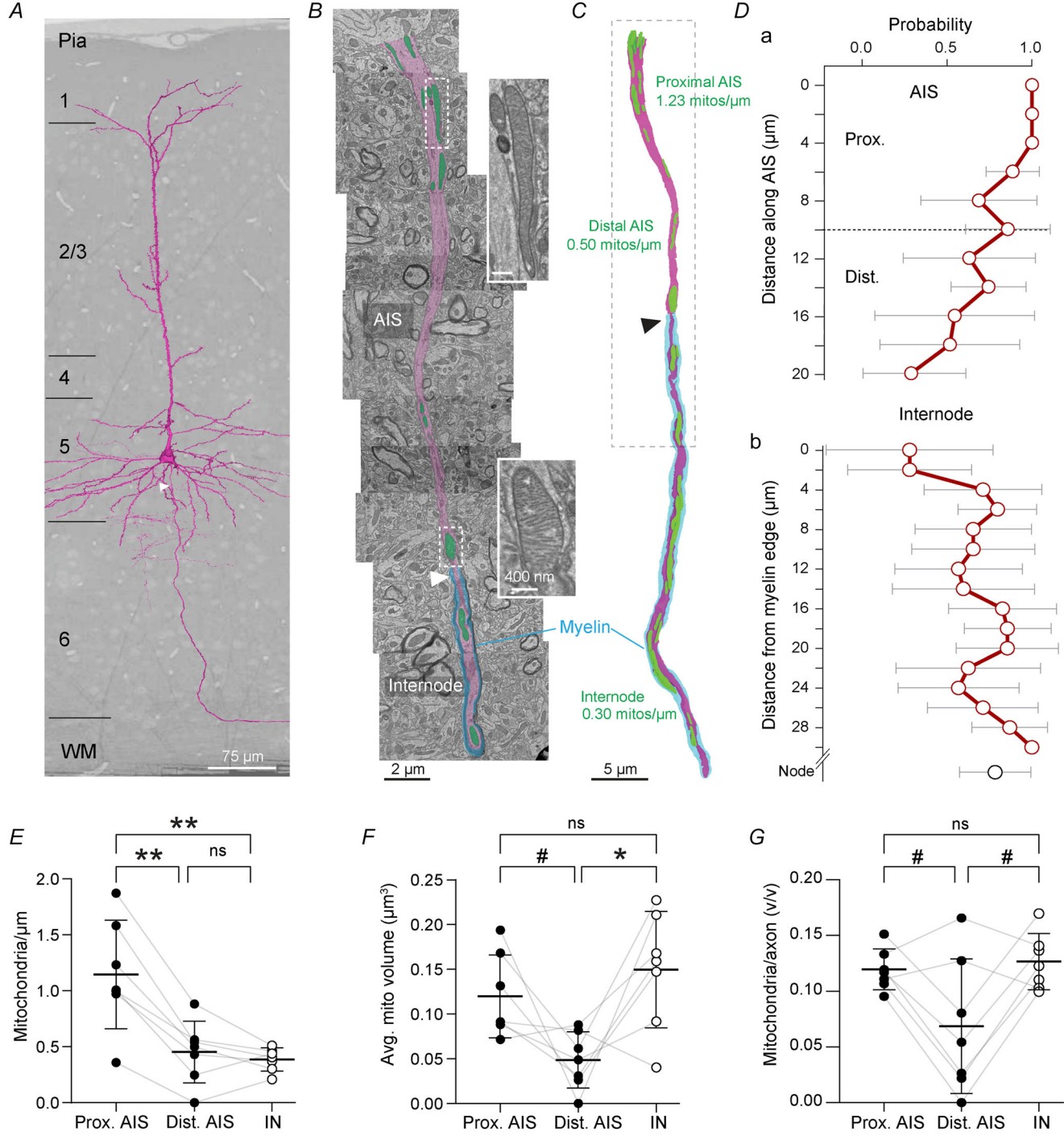

**Figure 1. Mitochondria densely populate the proximal axon initial segment (AIS)**

*A*, 3D reconstructed morphology of a thick-tufted layer 5 pyramidal neuron (magenta) projected on the electron microscopy (EM) block of a mouse visual cortex. White arrow, location of the axon. *B*, collage of single EM images of the AIS and a section of the first internode of the cell shown in *A*. AIS is pseudocoloured in magenta, mitochondria are pseudocoloured in green, myelin is pseudocoloured in cyan. Note that this is a single plane, underestimating the number of mitochondria. Dashed boxes correspond to example mitochondria. White arrowhead, paranodal loops of first internode. *C*, 3D reconstruction of the mitochondria and axon. Dashed box corresponds with the view in *A*, black arrowhead indicates paranodal loops of the first internode. *D*, distribution of mitochondria within the AIS (*Da,*) or internodes and nodes (*Db*), dashed line at 10 μm separating proximal *versus* distal AIS. *Y* axes are limited to lengths where each AIS/internode is represented. Error bars represent standard deviation. *E–G*, mitochondria are more densely populated at the proximal AIS compared with the distal AIS and first internode (*E*, repeated measures one-way ANOVA *P* = 0.0011; Tukey's *post hoc* test prox *vs.* dist AIS **P = 0.0011; Prox AIS *vs.* internode

**P = 0.0071; Dist AIS *vs*. internode P = 0.6973). At the distal AIS, a trend of smaller mitochondria is observed both in absolute (*F*, repeated measures one-way ANOVA P = 0.0132; Tukey's *post hoc* test prox *vs*. dist AIS P = 0.0828; Prox AIS *vs*. internode P = 0.3109; Dist AIS *vs*. internode P = 0.0419) and relative volume (*G*, repeated measures one-way ANOVA P = 0.0369; Tukey's *post hoc* test prox *vs*. dist AIS P = 0.0963; Prox AIS *vs*. internode P = 0.3434; Dist AIS *vs*. internode P = 0.0797). *D–G*, n = 7 axons from one mouse.

our findings from EM, immunohistochemistry and live imaging indicate that mitochondria localize mostly to the proximal AIS and are stable.

### Mitochondria at the AIS buffer Ca$^{2+}$

The AIS and nodes of Ranvier of layer 5 pyramidal neurons are characterized by large activity-dependent Ca$^{2+}$ fluxes (Hanemaaijer et al., 2020). We therefore asked whether the activity-dependent Ca$^{2+}$ buffering of mitochondria at the AIS differed from those in nodes, internodes and somata and how it correlated with cytosolic Ca$^{2+}$ influx. To this end we expressed the genetically encoded, GFP-based and mitochondrion-targeted Ca$^{2+}$ indicator mt-GCaMP6f selectively in L5 pyramidal neurons (Kole et al., 2022). We then performed whole-cell

patch-clamp recordings in mt-GCaMP6f-expressing (Kd ∼375 nM; Chen et al.) L5 pyramidal neurons with intracellular solution supplemented with the red Ca$^{2+}$ indicator dye Cal-590 (Kd 561 nM) (Fig. 4*A*). The dual Ca$^{2+}$ indicator approach enabled us to image the activity-dependent Ca$^{2+}$ responses in the soma, AIS, myelinated internodes and nodes simultaneously in the cytosol (cyt-Ca$^{2+}$) and mitochondria (mt-Ca$^{2+}$). The data showed that during spike trains, mt-Ca$^{2+}$ transients temporally followed those in the cytosol (Fig. 4*B*). Indeed, in line with previous reports (Groten & MacVicar, 2022; Stoler et al., 2022), we found that mt-Ca$^{2+}$ increase was significantly delayed by ∼150 ms (time to 50% peak; mt-Ca$^{2+}$, 315.39 ± 18.37 ms, cyt-Ca$^{2+}$, 149.43 ± 8.19 ms, paired *t* test P < 0.0001; n = 9 cells from six mice) and comparable decay time constant (mt-Ca$^{2+}$, 426.20

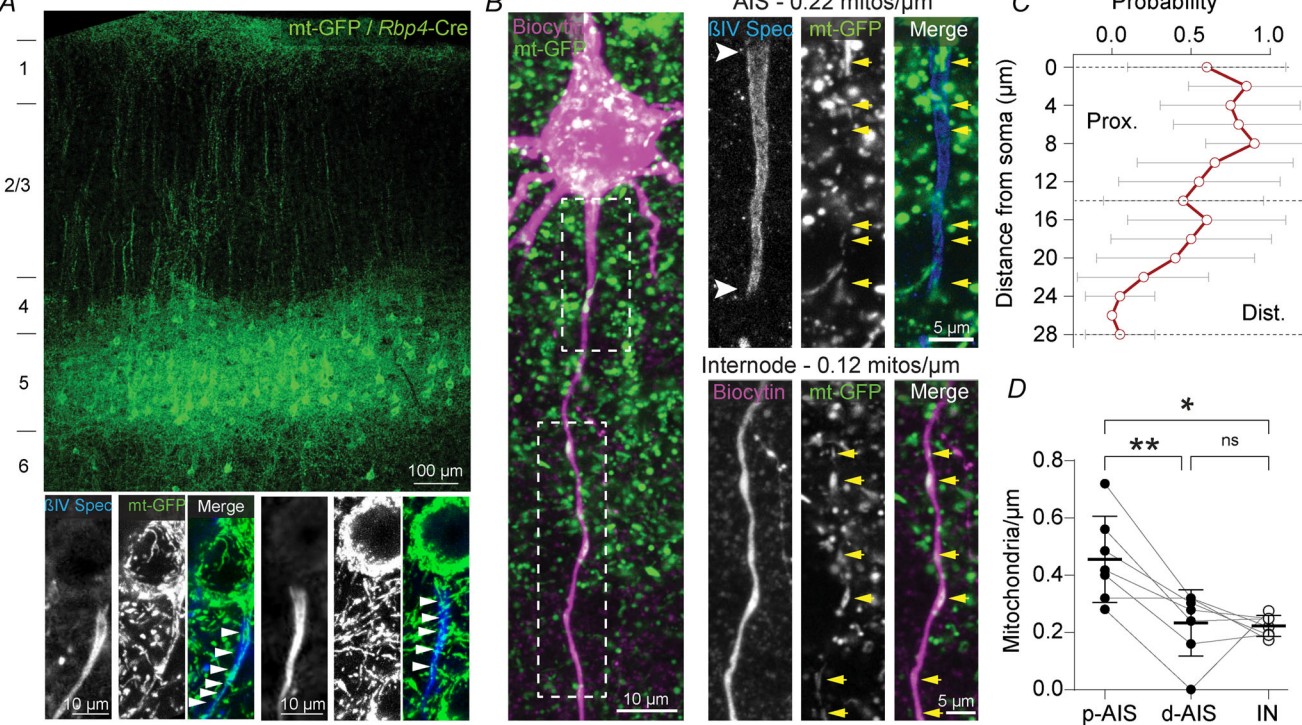

**Figure 2. Mitochondria densely populate the proximal axon initial segment (AIS)**
*A*, expression of mt-GFP in *Rbp4*-Cre mice labels mitochondria selectively in L5 pyramidal neurons. Insets show dense mitochondria at the AIS, visualized by $\beta$4 spectrin immunolabelling. *B*, example of a biocytin-filled mitoGFP$^+$ L5 pyramidal neuron. Dashed boxes correspond to the high magnification images on the right (AIS and first internode). White arrowheads indicate AIS, yellow arrows indicate mitochondria. *C*, mitochondria are most prevalent at the proximal AIS and reduce in density along the AIS length. *D*, mitochondrial density is highest at the proximal AIS (average 0.36 *vs*. 0.22 mitochondria/μm; repeated measures one-way ANOVA P = 0.0033; Tukey's *post hoc* test prox *vs*. dist AIS **P = 0.0097; Prox AIS *vs*. internode *P = 0.0282; Dist AIS *vs*. internode P = 0.9783); *C*, n = 20 AISs (not biocytin-filled) from two mice; *D*, n = 7 cells from five mice.

$\pm$ 36.3 ms; cyt-Ca$^{2+}$ 310.02 $\pm$ 57.15 ms, paired $t$ test $P = 0.0590$, $n = 9$ cells from six mice). It should be noted that we underestimate the decay duration, as our imaging sessions only captured the first few seconds of decay which in mitochondria can last for tens of seconds (Groten & MacVicar, 2022; Stoler et al., 2022).

We then compared cyt-Ca$^{2+}$ and mt-Ca$^{2+}$ response amplitudes from the axonal subcompartments, which revealed that both cyt-Ca$^{2+}$ and mt-Ca$^{2+}$ responses were strongest in the AIS whereas they were very weak to non-existent in myelinated internodes (Fig. 4$B$–$E$), in agreement with previous observations (Hanemaaijer et al., 2020; Kole et al., 2022). In nodes we detected AP-dependent cyt-Ca$^{2+}$ increase, but these were irregularly paired with mt-Ca$^{2+}$ transients (Fig. 4$C$–$D$), suggesting that the Ca$^{2+}$ influx may be insufficient for MCU activation. Given the variance in mitochondrial size (Fig. 1$C$), we asked whether mt-Ca$^{2+}$ responses varied by size and correlated mt-Ca$^{2+}$ peak responses with surface area. This revealed that smaller mitochondria displayed larger Ca$^{2+}$ transients (Fig. 4$F$). We did not observe a correlation between mt-Ca$^{2+}$ transient amplitude and distance from soma (Pearson correlation coefficient $R^2 = 0.0209$, $P = 0.4811$). Taken together, these findings

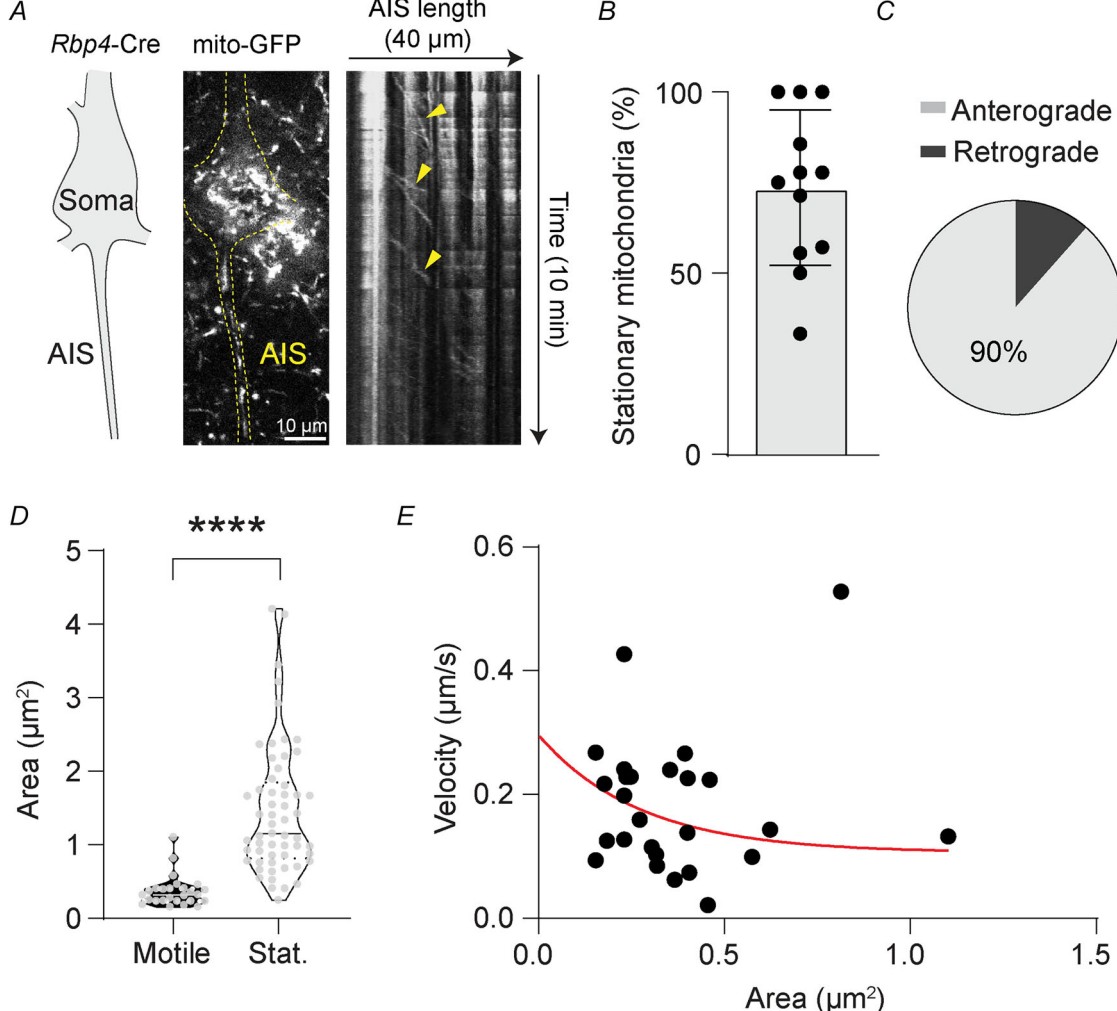

**Figure 3. Stable and motile mitochondria at the axon initial segment (AIS)**
*A*, example kymograph showing motility of some mitochondria (yellow arrows) but with most mitochondria remaining stably present during the imaging session. Somata are readily recognized by the density of mitochondria (see also 1*A*). In this example, the soma was on a different *z*-plane from the AIS. *B*, quantification of stationary mitochondria at the AIS. *C*, quantification of anterograde and retrograde movement by mitochondria at the AIS. *D*, stationary mitochondria are larger than motile mitochondria (nested $t$ test, ****$P < 0.0001$). *E*, no correlation between mitochondrial surface area and velocity (Pearson's $r = -0.08265$, $P = 0.6881$). Red line indicates an exponential decay ($y = 0.1871 \times \exp(-3.645 \times x) + 0.1086$). In *B*, each datapoint represents one AIS ($n = 12$ AISs from five mice); in *D* and *E*, each datapoint represents one mitochondrion (*D*, $n = 27$ motile and 57 stationary mitochondria from five AISs from five mice; *E*, $n = 27$ mitochondria from five AISs from five mice).

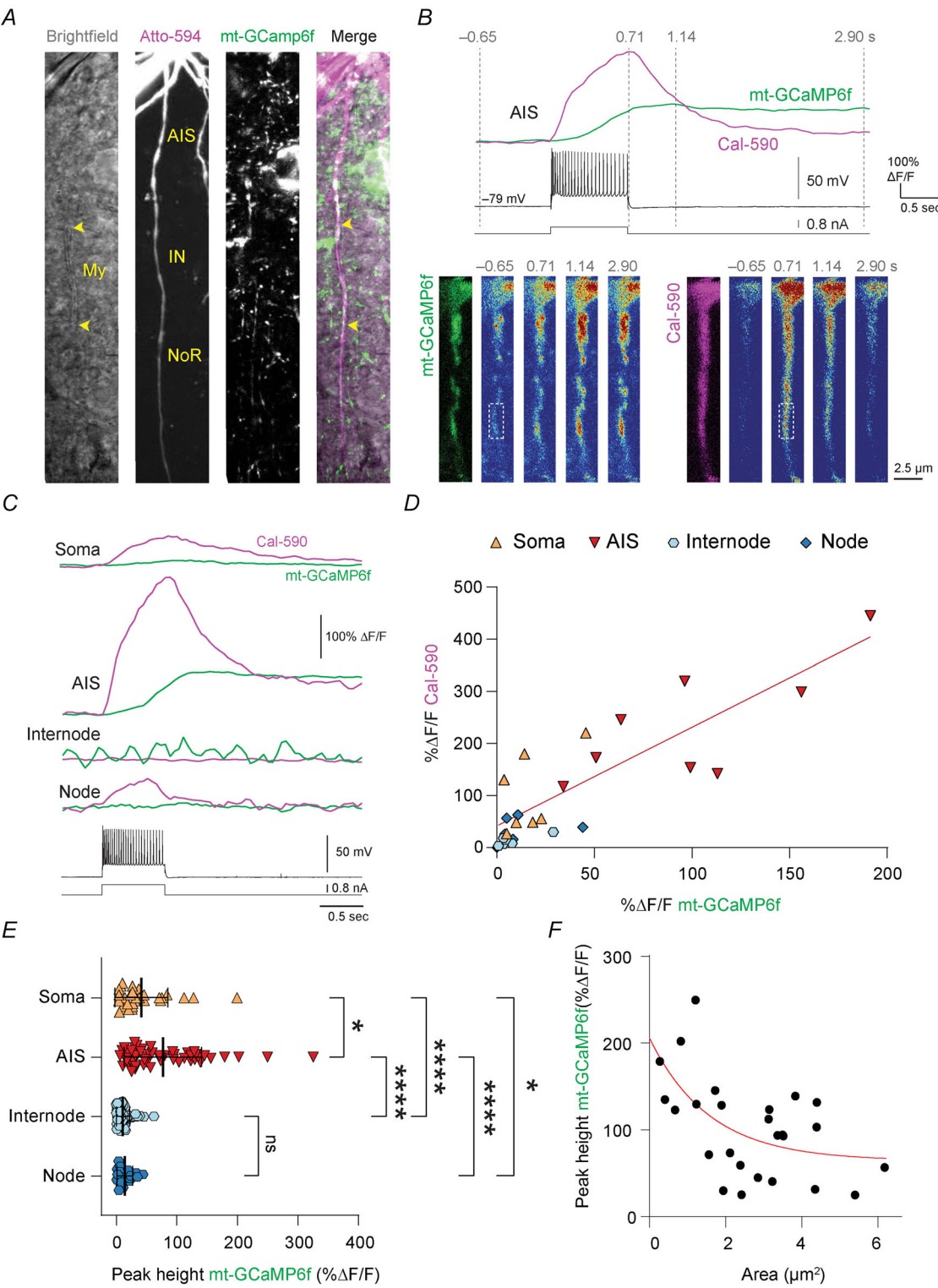

**Figure 4. Activity-dependent cyt-Ca²⁺ and mt-Ca²⁺ dynamics at the axon initial segment AIS**

*A*, example image of a mt-GCaMP6f⁺ L5 pyramidal neuron filled with Atto594 revealing the cytoarchitecture while transmission light reveals myelinated internodes (My, yellow arrowheads). *B, top*, example traces of mitochondrial (mt-GCaMP6f, green) and cytosolic (Cal-590, magenta) Ca²⁺ transients in response to a 700 ms AP train (25 APs). *Bottom*, heatmaps of the corresponding traces at the indicated timepoints. Note the difference in onset and duration between mt- and cyt-Ca²⁺ responses. Dashed boxes indicate regions of interest corresponding to

traces. Scale bar represents 2.5 μm. *C*, example traces from different cellular compartments from the same cell of mt- and cyt-Ca$^{2+}$ responses during a 700 ms AP train. *D*, scatter plot of mt- and cyt-Ca$^{2+}$ responses at soma, AIS, internode and nodes. The strongest responses are observed at the AIS. Dashed line indicates a linear curve fit ($y = 0.4096 + 1.896 \times x$). *E*, quantification of subcellular mt-Ca$^{2+}$ responses. Datapoints indicate somata or, in the case of AIS, internode and node, individual mitochondria (Kruskal–Wallis test with Dunn's *post hoc* test; AIS *vs.* Soma \*$P = 0.0481$, AIS *vs.* Internode \*\*\*\*$P < 0.0001$, AIS *vs.* Node \*\*\*\*$P < 0.0001$, Soma *vs.* Internode \*\*\*\*$P < 0.0001$, Soma *vs.* Node \*$P = 0.0417$, Internode *vs.* Node $P > 0.9999$). *F*, smaller mitochondria exhibit stronger Ca$^{2+}$ responses. Red line indicates a single exponential fit ($y = 1.4145 \times \exp(-0.6339 \times x) + 0.6435$). *D*, $n = 21$ cells from 12 mice; *E*, $n = 30$ somata, 55 (AIS), 83 (internode) and 19 (node) mitochondria from 40 cells of 21 mice; *F*, $n = 26$ mitochondria from eight AISs of six mice.

indicate that in L5 pyramidal neuron axons, mitochondria at the AIS strongly buffer spiking activity-dependent cytosolic Ca$^{2+}$.

## Mt-Ca$^{2+}$ buffering shortens the slow AHP duration

Mt-Ca$^{2+}$ buffering has previously been shown to attenuate cyt-Ca$^{2+}$ concentration and shorten the duration of the slow AHP (Groten & MacVicar, 2022). Since cyt-Ca$^{2+}$ at the AIS of L5 pyramidal neuron reaches high concentrations, shaping both single AP properties and the slow AHP (Hanemaaijer et al., 2020; Roshchin et al., 2020), we hypothesized that mt-Ca$^{2+}$ buffering in L5 pyramidal neurons affects the membrane excitability. To test this, we performed whole-cell patch-clamp recordings while infusing the selective MCU inhibitor Ru360, which blocks the MCU by binding it at the mitochondrial intermembrane space (Baughman et al., 2011), effectively abolishing mitochondrial Ca$^{2+}$ uptake. We observed that 45 min of Ru360 infusion attenuated mt-Ca$^{2+}$ with an average reduction of 81% compared with control cells (Fig. 5*A*–*D*; control, $2.74 \pm 2.06$; Ru360, $0.51 \pm 0.30$ $\Delta$F/F, $P = 0.0459$, unpaired *t* test). Next, we examined the role of blocking mt-Ca$^{2+}$ on membrane excitability using a variety of stimulation paradigms including those used previously (Groten & MacVicar, 2022; Kirchner et al., 2024). By injecting temporally precise brief currents we evoked 30 APs at 20 Hz, 200 APs at 20 Hz, 200 APs at 50 Hz or 100 APs at 100 Hz and quantified the slow AHP peak amplitude, half-width and area under the curve (AUC, Fig. 5*E*). In agreement with previous work (Groten & MacVicar, 2022), we found that Ru360 significantly increased the duration, but not the peak or the AUC of the AHP, selectively at high frequencies (50 or 100 Hz; 5*F* and *G*; unpaired *t* tests; peak $P = 0.9184$ and $P = 0.3354$, half-width, $P = 0.0255$ and $P = 0.0158$, AUC $P = 0.4057$ and $P = 0.2388$ for 50 Hz or 100 Hz, respectively). When we quantified the amplitude of the AHP at 2 s after AP stimulus termination there was no treatment effect detectable (unpaired *t* tests, $P = 0.6832$ and $P = 0.5958$ for 50 Hz and 100 Hz, respectively; membrane potential relative to rest: control 50 Hz, $-3.57 \pm 0.49$; control 100 Hz, $-2.13 \pm 0.69$; Ru360 50 Hz, $-3.45 \pm 0.64$ mV; Ru360 100 Hz, $-1.98 \pm 0.88$ mV). Furthermore, slower

membrane currents were not affected as the membrane potential returned to baseline at comparable rates between groups (time constant of one-phase exponential decay after 100 Hz stimulation; unpaired *t* test $P = 0.9099$; control, $10.54 \pm 4.96$; Ru360, $10.91 \pm 4.97$ s; $n = 5$ cells from three mice in each group). In line with this, the resting membrane potentials were not affected by Ru360 (unpaired *t* test, $P = 0.8599$; 100 Hz control, $-77.42 \pm 2.67$ mV; Ru360, $-77.14 \pm 3.23$ mV).

## AP properties are independent of mt-Ca$^{2+}$ buffering

Since the slow AHP in L5 pyramidal neurons was modulated in part by mt-Ca$^{2+}$ buffering we postulated that this could also be true for other calcium-sensing voltage-gated channels at the soma and AIS, impacting the firing properties. The slowly activating Kv7 channels, expressed at somatodendritic membrane and at high densities in the distal AIS, sets the neuronal resting membrane properties, regulates AP threshold and is gated by cytoplasmic Ca$^{2+}$ (Battefeld et al., 2014; Bernardo-Seisdedos et al., 2018; Delmas & Brown, 2005; Martinello et al., 2015). With the Ru360-mediated loss of mt-Ca$^{2+}$ buffering (Fig. 5) the putatively increased cytoplasmic Ca$^{2+}$ is expected to produce a closure of the Kv7 channel and thereby increase the neuronal excitability, by lowering the current threshold for AP generation. To explore this possibility, we injected incremental current steps and compared AP generation in control and Ru360 treated cells. Since mt-Ca$^{2+}$ buffering occurs relatively slow, with a ∼150 ms delay and peaking at ∼1 s during AP firing (Fig. 4*C*), the Ru360 treatment could preferentially impact the APs at the end of the train. To test this, we investigated the final AP of each train evoked during a 700 ms duration current injection (Fig. 6*A*). However, we found no effect of Ru360 on AP voltage threshold (control, $-50.79 \pm 2.64$ mV; Ru360, $-52.01 \pm 3.60$ mV; $P = 0.4517$, unpaired *t* test), amplitude (control, $78.61 \pm 4.10$ mV; Ru360, $80.34 \pm 3.46$ mV; $P = 0.1863$, unpaired *t* test) or half-width (control, $0.63 \pm 0.06$ ms; Ru360, $0.64 \pm 0.06$ mV; $P = 0.8769$, unpaired *t* test). We also could not detect a difference between the number of APs generated upon current injection (Fig. 6*B*; two-way ANOVA; treatment effect $P = 0.8250$; interaction effect

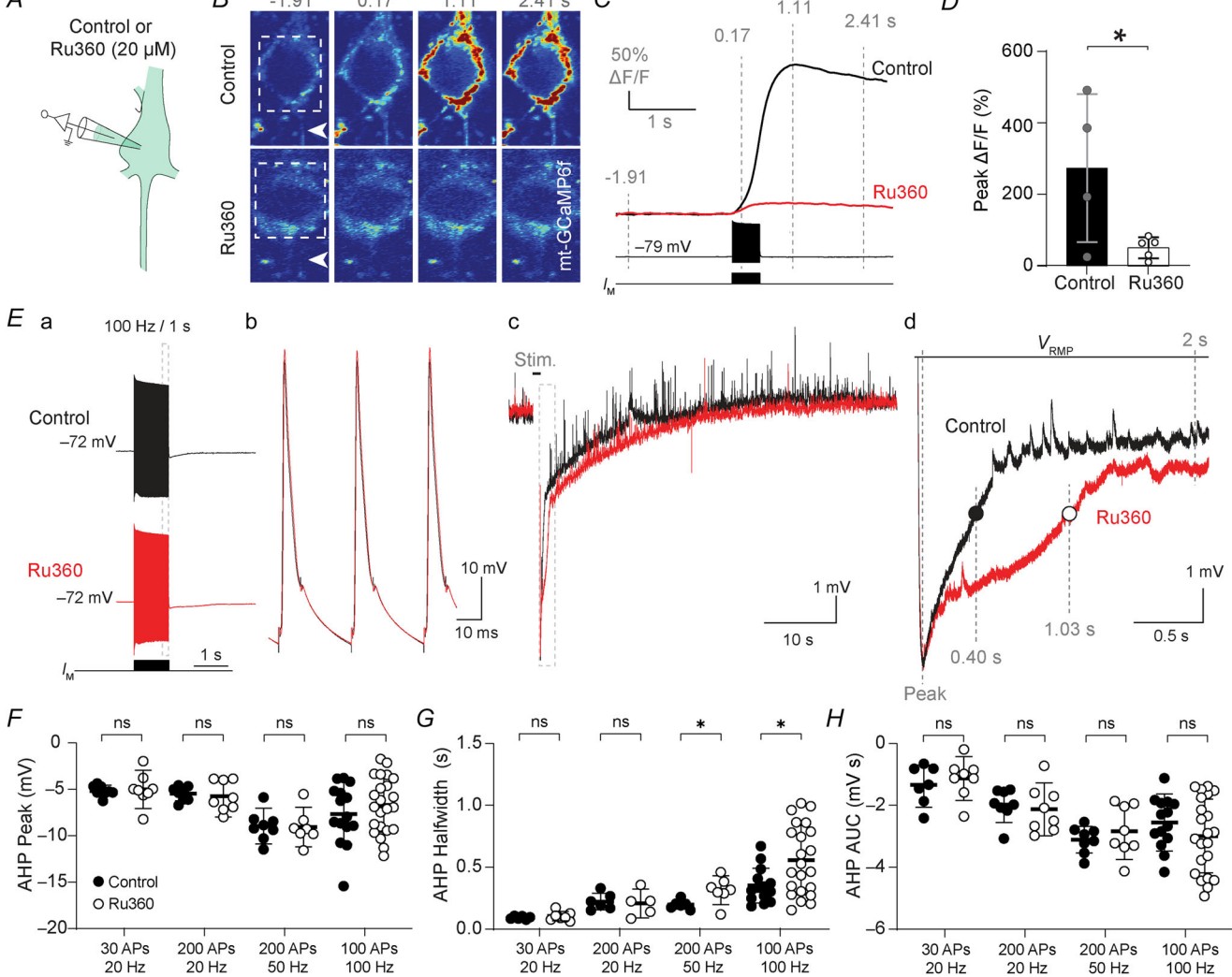

**Figure 5. mt-Ca²⁺ buffering reduces the high-frequency action potential (AP)-evoked slow after-hyperpolarization (AHP)**

*A*, schematic representation of the experiment. Whole-cell recordings are performed while infusing intracellular solution with or without Ru360. *B*, example mt-Ca²⁺ responses in a cell infused with intracellular without control (top) or with Ru360 (bottom). Arrowhead indicates proximal AIS; dashed box indicates the region of interest used for the traces in *C*. *C*, traces corresponding to the example cells in *B*. *D*, quantification of somatic mt-Ca²⁺ responses in control or Ru360-infused cells (unpaired *t* test, *$P$ = 0.0459). *Ea*, example voltage traces (100 APs, 100 Hz) of a control or Ru360-infused cell. Dashed box corresponds to final three APs; *Eb*, the final three APs from *Ea* at high temporal resolution. For clarity, capacitance transients are clipped; *Ec*, full trace of the AHP and return to baseline (full sweep length 60 s). Dashed box indicates the AHP. *Ed*, AHP at high temporal resolution. Notice the increased half-width. *F–H*, quantification of the AHP in response to the four tested stimulation paradigms. *F*, AHP peak amplitude (unpaired *t* tests; 30 APs 20 Hz $P$ = 0.8134; 200 APs 20 Hz $P$ = 0.6442; 200 APs 50 Hz $P$ = 0.9184; 100 APs 100 Hz $P$ = 0.3354). *G*, half-width (unpaired *t* tests; 30 APs 20 Hz $P$ = 0.4579; 200 APs 20 Hz $P$ = 0.3920; 200 APs 50 Hz *$P$ = 0.0255; 100 APs 100 Hz *$P$ = 0.0158). *H*, area under the curve (unpaired *t* tests; 30 APs 20 Hz $P$ = 0.5457; 200 APs 20 Hz $P$ = 0.6408; 200 APs 50 Hz $P$ = 0.4057; 100 APs 100 Hz $P$ = 0.9980). *D*, control, $n$ = 4 cells from two mice; Ru360, $n$ = 5 cells from two mice. *F*, 30 APs 20 Hz control, $n$ = 6 cells from three mice, Ru360, $n$ = 7 cells from four mice; 200 APs 20 Hz control, $n$ = 8 cells from five mice, Ru360, $n$ = 9 cells from five mice; 200 APs 50 Hz control, $n$ = 8 cells from five mice, Ru360, $n$ = 9 cells from five mice; 100 APs 100 Hz control, $n$ = 15 cells from 10 mice, Ru360, $n$ = 23 cells from 11 mice. *G*, 30 APs 20 Hz control, $n$ = 8 cells from five mice, Ru360, $n$ = 9 cells from four mice; 200 APs 20 Hz control, $n$ = 7 cells from five mice, Ru360, $n$ = 5 cells from four mice; 200 APs 50 Hz control, $n$ = 6 cells from four mice, Ru360, $n$ = 7 cells from four mice; 100 APs 100 Hz control, $n$ = 14 cells from nine mice, Ru360, $n$ = 22 cells from 10 mice. *H*, 30 APs 20 Hz control, $n$ = 7 cells from four mice, Ru360, $n$ = 8 cells from five mice; 200 APs 20 Hz control, $n$ = 8 cells from four mice, Ru360, $n$ = 8 cells from five mice; 200 APs 50 Hz control, $n$ = 8 cells from four mice, Ru360, $n$ = 8 cells from five mice; 100 APs 100 Hz control, $n$ = 14 cells from eight mice, Ru360, $n$ = 16 cells from eight mice.

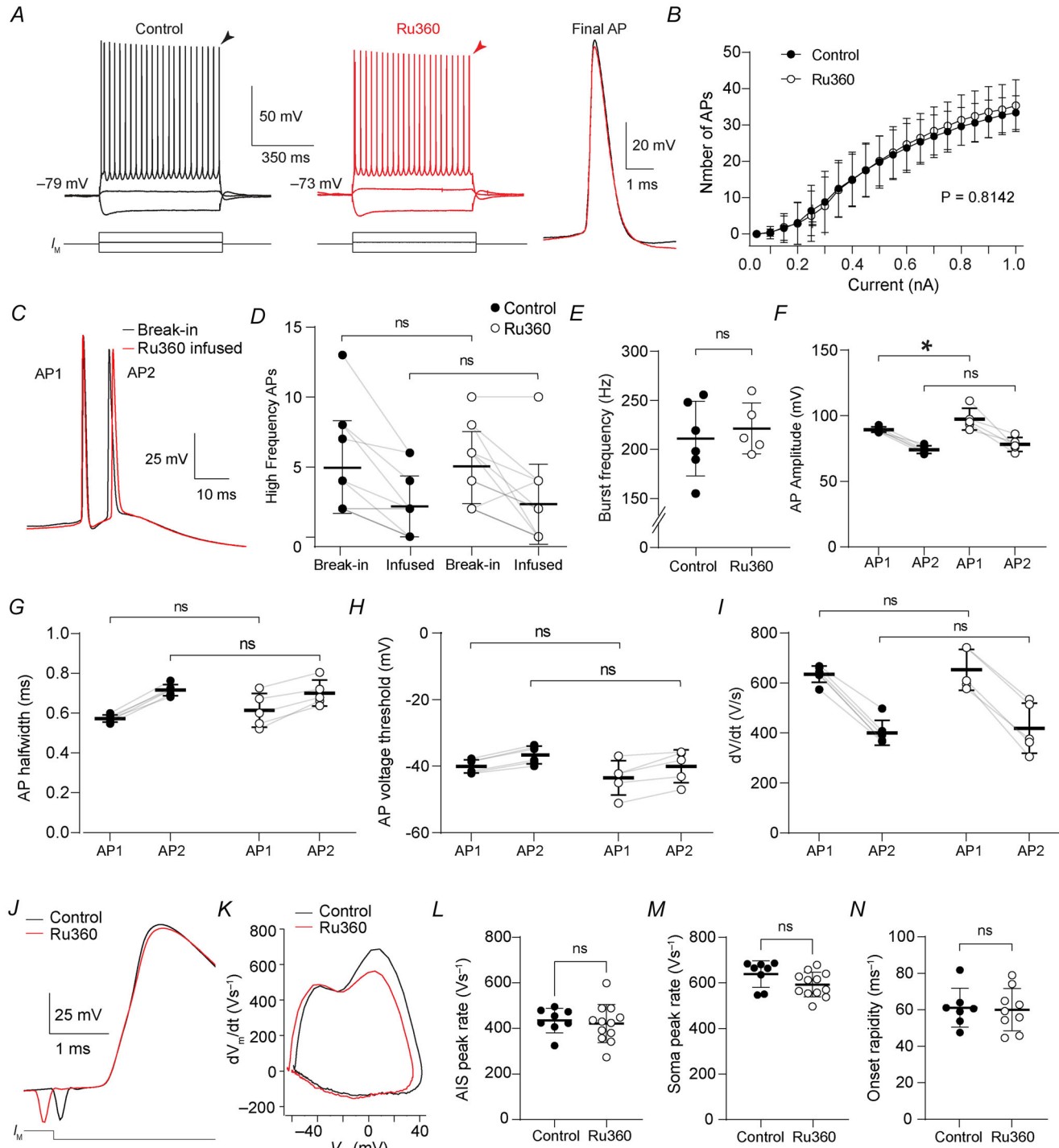

**Figure 6. mt-Ca²⁺ does not regulate AP initiation, waveform or bursting**

*A*, example traces of AP trains in a control or Ru360-infused cell. Arrowheads indicate the final APs shown in the inset on the right. *B*, population data showing unchanged firing frequency upon somatic current injection upon Ru360 infusion (two-way ANOVA; treatment effect $P = 0.8142$, $F_{(1, 17)} = 0.05041$; interaction effect $P = 0.8250$, $F_{(19, 323)} = 0.7046$; uncorrected Fisher's *post hoc* test, control *vs.* Ru360 $P = 0.9413$ at break-in and $P = 0.9024$ after infusion). *C*, example traces of high-frequency burst of the same cell at break-in and after Ru360 infusion. *D*, quantification of the number of high-frequency APs at break-in or after control or Ru360 infusion (two-way ANOVA; treatment effect $P = 0.9145$, $F_{(1, 21)} = 0.01181$; interaction effect $P = 0.9535$, $F_{(1, 21)} = 0.0003$). *E*, quantification of the within-burst frequency shows no effect of mt-Ca²⁺ buffering inhibition (unpaired *t* test,

$P = 0.6201$). *F–I*, quantification of within-burst AP amplitude (*F*, two-way ANOVA; treatment effect $P = 0.0201$, $F_{(1, 9)} = 7.948$; interaction effect $P = 0.3967$, $F_{(1, 9)} = 0.7920$; uncorrected Fisher's *post hoc* test, Control *vs.* Ru360, AP1 *$P = 0.0184$, AP2 $P = 0.2071$; AP1 *vs.* AP2 Control $P = 0.0006$, Ru360 $P = 0.0002$), half-width (*G*, two-way ANOVA; treatment effect $P = 0.6952$, $F_{(1, 9)} = 0.1638$; interaction effect $P = 0.0076$, $F_{(1, 9)} = 11.72$; uncorrected Fisher's *post hoc* test, Control *vs.* Ru360, AP1 $P = 0.2291$, AP2 $P = 0.6502$; AP1 *vs.* AP2 Control and Ru360 $P < 0.0001$), voltage threshold (*H*, two-way ANOVA; treatment effect $P = 0.1651$, $F_{(1, 9)} = 2.283$; interaction effect $P = 0.9941$, $F_{(1, 9)} = 5.784 \times 10^{-5}$; uncorrected Fisher's *post hoc* test, Control *vs.* Ru360, AP1 $P = 0.1556$, AP2 $P = 0.1549$; AP1 *vs.* AP2 Control $P = 0.0002$, Ru360 $P = 0.0004$) and d*V*/d*t* (*I*, two-way ANOVA; treatment effect $P = 0.6699$, $F_{(1, 9)} = 0.1942$; interaction effect $P = 0.9877$, $F_{(1, 9)} = 0.0002518$; uncorrected Fisher's *post hoc* test, Control *vs.* Ru360, AP1 $P = 0.6794$; AP2 $P = 0.6733$; AP1 *vs.* AP2 Control and Ru360 $P < 0.0001$). *J*, example traces of single APs (aligned at AP onset) elicited by a 3 ms 1 nA pulse in a control and a Ru360-infused cell. *K*, example phase plane plots of the APs in *J*. *L–N*, population data showing unchanged AIS component peak (*L*, unpaired *t* test, $P = 0.6935$), somatic component peak (*M*, unpaired *t* test, $P = 0.0856$) and rapidity of AP onset (*L*, unpaired *t* test, $P = 0.8543$). *B*, control, $n = 10$ cells from four mice; Ru360, $n = 10$ cells from four mice. *D*, control, $n = 11$ cells from six mice; Ru360, $n = 12$ cells from six mice. *E*, control, $n = 6$ cells from four mice; Ru360, $n = 5$ cells from four mice. *F–I*, control, $n = 6$ cells from four mice; Ru360, $n = 5$ cells from four mice. *L*, control, $n = 8$ cells from four mice; Ru360, $n = 9$ cells from three mice. *L–M*, control, $n = 8$ cells from four mice; Ru360, $n = 12$ cells from five mice. *N*, control, $n = 7$ cells from three mice; Ru360, $n = 9$ cells from three mice.

$P = 0.8142$), neither a change of the rheobase current (unpaired *t* test, $P = 0.3355$; control, $244.4 \pm 84.57$; Ru360, *vs.* $281.3 \pm 65.12$ pA) nor a change of the input resistance (unpaired *t* test, $P = 0.2820$; control, $47.04 \pm 11.70$ pA; Ru360, *vs.* $40.87 \pm 8.57$ pA).

L5 pyramidal neurons are characterized by their ability to generate short high-frequency (>100 Hz) bursts, which predominantly occur during onset of membrane depolarization. Burst firing is mediated by the rapid opening of Na$^+$ channels and shaped by both the Kv7 and BK channels (Battefeld et al., 2014; Niday & Bean, 2021; Roshchin et al., 2018) which are all Ca$^{2+}$-gated (Ben-Johny et al., 2014; Bernardo-Seisdedos et al., 2018; Niday & Bean, 2021; Roshchin et al., 2018; Wang et al., 2014). We postulated that mt-Ca$^{2+}$ regulation of membrane excitability may occur at the timescale of short inter-spike intervals and therefore its contribution could be masked when quantifying average rates. We therefore investigated the role of mt-Ca$^{2+}$ buffering to burst properties. First, we compared the number of high-frequency APs at rheobase immediately after establishing whole-cell access with the number after 45 min of Ru360 or control infusion. In both the control and Ru360-infused neurons, we observed a reduction in the number of high-frequency APs, likely due to washout of the cytoplasm with the intracellular recording solution (Fig. 6*D*; two-way ANOVA; infusion effect $P < 0.0001$, $F_{(1, 21)} = 27.64$; uncorrected Fisher's *post hoc* test, break-in *vs.* infusion, control $P = 0.0014$, Ru360 $P = 0.0012$). This reduction was, however, indistinguishable between the two treatments (two-way ANOVA with uncorrected Fisher's *post hoc* test, control *vs.* Ru360 $P = 0.9413$ at break-in and $P = 0.9024$ after infusion). Similarly, the fraction of neurons that maintained burst firing was similar between the two groups (control, 6 out of 9 or 66.7%; Ru360, 6 out of 11 or 54.5%; $P = 0.5231$, binomial test). Secondly, in

the stable burst firing neurons we compared the AP properties within the burst cluster. Quantification of the burst frequency showed no difference upon mt-Ca$^{2+}$ buffering inhibition (Fig. 6*E*; control, $211.0 \pm 38.0$ Hz; Ru360, $221.3 \pm 26.1$ Hz; unpaired *t* test $P = 0.6201$). While Ru360 infusion increased the amplitude for the first AP within a burst cluster (Fig. 6*F*; two-way ANOVA with uncorrected Fisher's *post hoc* test, $P = 0.0184$; control, $89.44 \pm 2.21$ mV; Ru360, $97.42 \pm 8.41$ mV) the amplitude of the second AP was unaffected (two-way ANOVA with uncorrected Fisher's *post hoc* test, $P = 0.2071$; control, $74.48 \pm 2.87$ mV; Ru360, $78.20 \pm 5.42$ mV). Furthermore, all AP voltage thresholds, half-widths and d*V*/d*t* values were comparable between groups (Fig. 6*G–I*; voltage threshold AP1 $P = 0.1556$, AP2 $P = 0.1549$; half-width AP1 $P = 0.2291$, AP2 $P = 0.6502$; d*V*/d*t* AP1 $P = 0.6794$ and AP2 $P = 0.6733$). Finally, to investigate whether mt-Ca$^{2+}$ buffering affects the AIS component of the AP, we recorded the upstroke of single APs (generated by a 3 ms pulse, 6*J–N*). Upon Ru360 infusion, the AP amplitude remained constant (unpaired *t* test, $P = 0.8359$), as well as the half-width (unpaired *t* test, $P = 0.5073$) and the current and voltage threshold (unpaired *t* tests; respectively $P = 0.5330$ and $P = 0.7096$). Furthermore, we made phase plane plots revealing clear voltage–time separation of the AIS and soma in the somatically recorded AP waveform. The results showed that Ru360 did not change the rate of rise peak of either the AIS or somatic component (Fig. 6*K–M*; AIS, unpaired *t* test, $P = 0.6935$; Soma, unpaired *t* test, $P = 0.0856$). In line with a lack of an effect on the AIS excitability, the rapidness of AP onset was unchanged upon Ru360 infusion (Fig. 6*N*; unpaired *t* test, $P = 0.8543$).

In conclusion, these electrophysiological experiments indicate that the large mt-Ca$^{2+}$ buffering at the AIS does not influence AP generation.

## Discussion

Combining cell-type specific ultrastructural data and genetically encoded $Ca^{2+}$ imaging tools we found that mitochondria at the AIS in neocortical L5 pyramidal neurons densely populate the proximal domain of the AIS and powerfully take up $Ca^{2+}$. AP generation, which is well established to produce some of the largest rises of cytoplasmic $Ca^{2+}$ within the AIS compartment (Bender & Trussell, 2009; Filipis et al., 2023; Hanemaaijer et al., 2020), was also associated with a substantial and compartmentalized mt-$Ca^{2+}$ uptake. Furthermore, in accord with Groten & MacVicar (2022) and Kirchner et al. (2024) our data showed that with high firing frequencies ($\geq$50 Hz) mt-$Ca^{2+}$ influx shortens the duration of the initial component of the slow AHP. Unexpectedly, despite the significant increase of the slow AHP with the use of the selective MCU inhibitor Ru360, the neuronal mt-$Ca^{2+}$ did not play a role in AP initiation or adaptation. Cytoplasmic $Ca^{2+}$ levels at the AIS regulates AP repolarization and width, as well as the inter-spike intervals (Bender & Trussell, 2009; Filipis et al., 2023; Gründemann & Clark, 2015; Hanemaaijer et al., 2020; Schwindt et al., 1992). Our analysis of somatically recorded APs, which reflects both AIS and somatodendritic excitability, revealed no coherent evidence that the kinetics and threshold properties of the AP were changed upon MCU block. This was true for single APs as well as the final AP in a train, which temporally coincides with significant mt-$Ca^{2+}$ uptake (Fig. 4C). In addition, the AIS component of the AP, indirectly visible in the phase plane plots, did not reveal any effect of Ru360-mediated mt-$Ca^{2+}$ block.

The lack of an effect of Ru360 on AP initiation may be related to the spatiotemporal features of spike generation and compartmentalization of the excitability. The slow AHP in L5 pyramidal neurons is thought to regulate spike-frequency adaptation and is detectable as an outward $K^+$ current mediated by the $Ca^{2+}$-dependent $K^+$ channel isoform $KCa_{3.1}$, lasting tens of seconds following a train of high-frequency spikes (Groten & MacVicar, 2022; Guan et al., 2015; Roshchin et al., 2020; Schwindt et al., 1992). The expression of $KCa_{3.1}$ is, however, restricted to the somatodendritic domain and $Ca^{2+}$ uncaging at the axon does not produce these slow outward currents (Roshchin et al., 2020), suggesting that the slow AHP is generated by the somatodendritic membrane and more likely shaped by mitochondria within the soma (Groten & MacVicar, 2022; Kirchner et al., 2024). Furthermore, the contribution of mt-$Ca^{2+}$ to the free $Ca^{2+}$ near the AIS plasma membrane could be too slow or masked by other $Ca^{2+}$ buffering mechanisms. Selectively in the thick-tufted L5 pyramidal neurons of the primary somatosensory cortex, the AIS harbours the giant saccular organelle, a large and specialized variant of the cisternal organelle consisting of smooth endoplasmic reticulum (ER), which

determines $\sim$50% of the AP-dependent $Ca^{2+}$ rise by $Ca^{2+}$-induced $Ca^{2+}$ release (Antón-Fernández et al., 2015; Carvalhais et al., 2026; Hanemaaijer et al., 2020). Although genetic ablation of the cisternal organelle does not impact AP firing rates (Orth et al., 2007), it remains a possibility that ER $Ca^{2+}$ handling interacts with mitochondria to shape cytoplasmic $Ca^{2+}$ at the AIS.

$Ca^+$ activated BK channels, which produce a fast AHP within milliseconds, are activated by $Ca^{2+}$ domains near the plasma membrane and possess sub-millisecond activation kinetics (Filipis et al., 2023; Niday & Bean, 2021). In comparison, we found that AIS mt-$Ca^{2+}$ buffering follows cyt-$Ca^{2+}$ transients with a $\sim$150 ms temporal delay, in good agreement with previous reports (Ashrafi et al., 2020; Groten & MacVicar, 2022; Stoler et al., 2022). This delay temporally overlaps with $Ca^{2+}$-gated Kv7 channels, which are partially open at rest and require tens of milliseconds to fully activate and regulate spike-frequency adaptation (Battefeld et al., 2014; Brown & Passmore, 2009). However, neither steady-state AP firing nor the rheobase were affected, suggesting that Kv7 channels are likely not modified by mt-$Ca^{2+}$ uptake (Fig. 5). An interesting alternative experimental approach for future studies would be to combine patch-clamp recordings with optical approaches to locally disrupt AIS mt-$Ca^{2+}$ buffering (Tjiang & Zempel, 2022; Tkatch et al., 2017). Although currently available tools do not specifically target mt-$Ca^{2+}$ uptake, an exciting possibility could be to develop light-mediated disruption of, for example, the MCU (Hermann et al., 2015). Taken together, based on the present lines of electrophysiological analyses, our data indicate that mt-$Ca^{2+}$ sequestration buffering does not play a role in the baseline AP properties of L5 pyramidal neurons. The finding that AP initiation and spike adaptation were independent from the MCU-associated $Ca^{2+}$ uptake is consistent with the generally preserved cognitive and motor control under baseline conditions in the MCU knockout mice (Nichols et al., 2016; Pan et al., 2013; Szibor et al., 2020).

### Anatomical distribution and non-electrical roles of AIS mitochondria

The low number of mitochondria at the distal AIS relative to the internodes (Figs 1 and 2) is in good support of recent emerging insights into their distribution in mammalian motor neurons, human-induced pluripotent stem cells and *Drosophila* AIS (Tamada et al., 2021; Tjiang & Zempel, 2022; Wodrich et al., 2024). Here, we extend these observations by identifying proximal–distal gradients of mitochondria, with significantly smaller ones in the distal domain of the AIS. The molecular mechanisms determining the mitochondria distribution along the AIS subdomains are not well understood.

$Ca^{2+}$-dependent anchoring of mitochondria is in part mediated by the adaptor protein Miro or the anchoring protein syntaphilin (Devine & Kittler, 2018). However, since $Ca^{2+}$ influx is uniformly high along the AIS and mitochondrial clustering also occurs in the absence of $Ca^{2+}$ influx, as shown in basket cell internodes (Kole et al., 2022) this mechanism is not very likely to account for the gradient. Another possibility is that mitochondria are tethered to the ER, for instance via vesicle-associated membrane protein-associated proteins (Bapat et al., 2024) or the axonal membrane (Lackner et al., 2013). To our knowledge, mitochondria are not physically tethered to voltage-gated ion channels but given their spatially non-uniform distributions along the proximal–distal axis (Jenkins & Bender, 2024; Kole & Stuart, 2012) it is interesting to speculate that mechanisms defining subdomain localization may be (partially) shared or that the ion channel distribution guides mitochondrial clustering.

Since AP initiation typically emerges from the distal AIS region (Jenkins & Bender, 2024; Kole & Stuart, 2012), our anatomical and functional data suggest that under baseline conditions mitochondrial $Ca^{2+}$ handling is dispensable for AP initiation. If excitability is not affected, what could be the role of mitochondria in the AIS? One intriguing possibility is that the MCU-mediated $Ca^{2+}$ importing at the AIS may be involved during prolonged periods of elevated cytoplasmic $Ca^{2+}$ and acts as an instructive signal for cellular homeostasis and structural plasticity, analogous to their role at spines and presynaptic terminals (Kwon et al., 2016; Lewis et al., 2018). In line with this idea, global MCU knockout preserves baseline activity levels and mitochondrial ultrastructure but renders cortical neurons vulnerable as the respiratory capacity fails to offset the increased glycolytic demands (Nichols et al., 2016; Szibor et al., 2020). During strongly increased levels of neuronal activity, the distal AIS shortens via endocytosis of voltage-gated $Na^+$ channels and Ankyrin G proteins from the distal AIS plasma membrane within a period of tens of minutes (Fréal et al., 2023; Jamann et al., 2021). AIS plasticity induced by neuronal activity or brief *N*-methyl-D-aspartate receptor activation is a $Ca^{2+}$-dependent process (Evans et al., 2013; Fréal et al., 2023). More extreme AIS plasticity induction protocols showed that AIS disassembly requires proteasomes. It is likely that mitochondria dock to sites of the AIS cytoskeleton during remodelling. An interesting experiment would be to genetically silence the MCU and image mt-GFP during AIS plasticity in cortical L2/3 cells (which display structural AIS plasticity under normal conditions, in contrast to their L5 counterparts (Jamann et al., 2021)).

Mitochondria at the AIS may play important roles in maintaining axonal integrity and become critical during pathological conditions such as demyelination and/or neurodegenerative diseases. In support of this view, the organelles increase in number and size within the AIS and accumulate at the distal AIS in injury models and amyotrophic lateral sclerosis (Sasaki & Iwata, 1996; Tamada et al., 2021; Wodrich et al., 2024). In demyelinated axons the mitochondrial anchoring protein syntaphilin is required for the immobilization, clustering and the increase of mitochondria size (Ohno et al., 2014). AIS mitochondria under pathological conditions may support the adaptation of neuronal excitability or direct axonal cargo transport. For example, motor neurons in amyotrophic lateral sclerosis accumulate mitochondria at the AIS and are characterized by a shorter AIS length and hypoexcitability (Harley et al., 2023; Sasaki & Iwata, 1996). Nerve injury normally induces proteasome-mediated disassembly of the AIS and subsequent mitochondrial transport to the distal axon to support injury repair. Interestingly, this process is disrupted in an amyotrophic lateral sclerosis mouse model (Kiryu-Seo et al., 2022). Mitochondria at the AIS have previously been described to control trafficking of TAU protein, suggesting that transport of other axonal cargos may also be regulated by mitochondria at the AIS (Tjiang & Zempel, 2022). The complex and bidirectional interactions between AIS integrity, mitochondria and axonal injury remain to be further studied. Pharmacologically enhancing mt-$Ca^{2+}$ buffering in hippocampal neurons rescues epilepsy symptoms, in which mt-$Ca^{2+}$ buffering at the AIS could be involved (Styr et al., 2019). Taken together, the current findings show that L5 pyramidal neurons regulate their axonal mitochondrial content at the microdomain level, that mitochondria cluster to the proximal AIS but are dispensable for maintaining normal levels of membrane excitability. These data open avenues to explore the roles of mitochondria in the (dis)assembly and maintenance of the AIS during plasticity and pathology.

# Appendix

**Table A1. Overview of antibodies and dilutions used in this study**

| Antibody | Host | Dilution | Manufacturer | Cat. number | RRID | Primary antibody validation |
|---|---|---|---|---|---|---|
| Anti-GFP | Chicken | 1:500 | Abcam | ab13970 | AB_300 798 | (Berkovits & Mayr, 2015; Blin et al., 2026) |
| Anti-$\beta$IV Spectrin | Rabbit | 1:1000 | Engelhardt and Kole lab | N/A | N/A | (Jamann et al., 2021) |
| Anti-Chicken-Alexa-488 | Goat | 1:500 | ThermoFisher | A11039 | AB_142 924 | N/A |
| Anti-Rabbit-Alexa-633 | Goat | 1:500 | ThermoFisher | A21070 | AB_2 535 731 | N/A |
| Streptavidin-Alexa-594 | N/A | 1:500 | ThermoFisher | S11227 | N/A | N/A |

**Table A2. Overview of the cells used in the 3D EM analysis. Cell L5PN_7 was used as the example in Fig. 1**

| Cell name | Neuroglancer ID | Soma coordinates |
|---|---|---|
| L5PN_1 | 864 691 135 280 056 225 | 181 036, 189 486, 20 898 |
| L5PN_2 | 864 691 135 463 505 821 | 204 687, 202 669, 20 853 |
| L5PN_3 | 864 691 135 293 275 574 | 201 708, 198 355, 21 239 |
| L5PN_4 | 864 691 136 990 817 685 | 227 311, 197 012, 21 697 |
| L5PN_5 | 864 691 135 384 023 787 | 209 082, 201 077, 21 549 |
| L5PN_6 | 864 691 135 725 697 451 | 219 205, 190 682, 22 103 |
| L5PN_7 | 864 691 135 408 575 689 | 214 530, 192 267, 22 033 |

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

## Additional information

### Data availability statement

All data are available upon request to the corresponding authors. The Microns dataset used for 3D EM analysis is available at www.microns-explorer.org. The script used to analyse $Ca^{2+}$ imaging data is available via https://github.com/Kolelab/mtGCaMP.

### Competing interests

The authors declare no conflicts of interest.

### Author contributions

This study was funded by a ZonMW Off Road grant 0 451 001 201 0066 to K.K. and a Netherlands Research Council (NWO) Vici grant 865.17.003 to M.K. The authors are indebted to the members of the Axonal Signalling department of the NIN for discussing preliminary data and manuscript versions, and Naomi Petersen for the support with AAV injections. K.K. and M.K. conceptualized the experiments, K.K. performed the experiments, K.K. and M.K. wrote and revised the manuscript. All authors approved the final version of the manuscript, agree to be accountable for all aspects of the work in ensuring that questions related to the accuracy or integrity of any part of the work are appropriately investigated and resolved. All persons designated as authors qualify for authorship, and all those who qualify for authorship are listed.

### Funding

ZonMw (Netherlands Organisation for Health Research and Development): Koen Kole, 0 451 001 201 0066; NWO | Exacte en Natuurwetenschappen (ENW): Maarten H.P. Kole, 865.17.003.

### Keywords

action potential, axon initial segment, calcium buffer, mitochondria, pyramidal neuron

## Supporting information

Additional supporting information can be found online in the Supporting Information section at the end of the HTML view of the article. Supporting information files available:

**Peer Review History**

