## [Peer Review History · The Journal of Physiology]

Robust activity-dependent mitochondrial calcium dynamics at the AIS is dispensable for action potential generation

Koen Kole and Maarten H.P. Kole

DOI: 10.1113/JP289290

Corresponding author(s): Maarten Kole (m.kole@nin.knaw.nl)

Review Timeline:

Submission Date:	24-May-2025
Editorial Decision:	08-Jul-2025
Revision Received:	14-Jan-2026
Editorial Decision:	05-Feb-2026
Revision Received:	07-Feb-2026
Accepted:	11-Feb-2026

Senior Editor: Nathan Schoppa

Reviewing Editor: Samuel Young

Transaction Report:

Dear Dr Kole,

Re: JP-RP-2025-289290 "Robust activity-dependent mitochondrial calcium dynamics at the AIS is dispensable for action potential generation" by Koen Kole and Maarten H.P. Kole

Thank you for submitting your manuscript to The Journal of Physiology. It has been assessed by a Reviewing Editor and by 2 expert referees and we are pleased to tell you that it is potentially acceptable for publication following satisfactory major revision.

REVISION CHECKLIST:

Please upload two versions of your manuscript text: one with all relevant changes highlighted and one clean version with no

changes tracked. The manuscript file should include all tables and figure legends, but each figure/graph should be uploaded as separate, high-resolution files.

We look forward to receiving your revised submission.

Yours sincerely,

Nathan Schoppa
Senior Editor
The Journal of Physiology

REQUIRED ITEMS

- Author photo and profile. First or joint first authors are asked to provide a short biography (no more than 100 words for one author or 150 words in total for joint first authors) and a portrait photograph. These should be uploaded and clearly labelled together in a Word document with the revised version of the manuscript. See Information for Authors for further details.
- Your manuscript must include a complete Additional Information section, including competing interests; funding; author contributions and acknowledgements.
- Please upload separate high-quality figure files via the submission form.
- Please ensure that the Article File you upload is a Word file.
- Your paper contains Supporting Information of a type that we no longer publish, including supplementary tables and figures. Any information essential to an understanding of the paper must be included as part of the main manuscript and figures. The only Supporting Information that we publish are video and audio, 3D structures, program codes and large data files. Your revised paper will be returned to you if it does not adhere to our Supporting Information Guidelines.
- Papers must comply with the Statistics Policy: https://jp.msubmit.net/cgi-bin/main.plex?form_type=display_requirements#statistics.

In summary:

- If $n \leq 30$, all data points must be plotted in the figure in a way that reveals their range and distribution. A bar graph with data points overlaid, a box and whisker plot or a violin plot (preferably with data points included) are acceptable formats.
- If $n > 30$, then the entire raw dataset must be made available either as supporting information, or hosted on a not-for-profit repository, e.g. FigShare, with access details provided in the manuscript.
- 'n' clearly defined (e.g. x cells from y slices in z animals) in the Methods. Authors should be mindful of pseudoreplication.
- All relevant 'n' values must be clearly stated in the main text, figures and tables.
- The most appropriate summary statistic (e.g. mean or median and standard deviation) must be used. Standard Error of the

Mean (SEM) alone is not permitted.

- Exact p values must be stated. Authors must not use 'greater than' or 'less than'. Exact p values must be stated to three significant figures even when 'no statistical significance' is claimed.

- Please include an Abstract Figure file, as well as the Figure Legend text within the main article file. The Abstract Figure is a piece of artwork designed to give readers an immediate understanding of the research and should summarise the main conclusions. If possible, the image should be easily 'readable' from left to right or top to bottom. It should show the physiological relevance of the manuscript so readers can assess the importance and content of its findings. Abstract Figures should not merely recapitulate other figures in the manuscript. Please try to keep the diagram as simple as possible and without superfluous information that may distract from the main conclusion(s). Abstract Figures must be provided by authors no later than the revised manuscript stage and should be uploaded as a separate file during online submission labelled as File Type 'Abstract Figure'. Please also ensure that you include the figure legend in the main article file. All Abstract Figures should be created using BioRender. Authors should use The Journal's premium BioRender account to export high-resolution images. Details on how to use and access the premium account are included as part of this email.

EDITOR COMMENTS

Reviewing Editor:

Comments for Authors to ensure the paper complies with the Statistics Policy (Required):
Data is presented as SEM, needs to be presented as SD

Comments to the Author (Required):

This manuscript focuses on mitochondria regulation in calcium dynamics at the AIS at layer 5 pyramidal neurons. Both reviewers found the conclusions largely supported by the data reported in the manuscript. However, both reviewers have concerns about the manuscript. Both reviewers point out that significant text revisions are needed to better explain logic of experiments and expand the discussion section in the article. In addition, reviewer#1 pointed out to strengthen conclusions there is a need to determine if different stimulation paradigms between the studies ci could explain the difference in results between the studies. Therefore, based on these positive comments and critiques address concerns in this manuscript an additional experiment that uses a similar stimulation paradigm as in previous studies to resolve what underpins the contrasting results is needed. Furthermore, a significant revision of the text will need to be made, and better discussion of the differences between this study and prior studies. Finally, the manuscript will need to comply with Journal policies for reporting of statistics as SD.

Please also see 'Required Items' above.

Senior Editor:

Comments for Authors to ensure the paper complies with the Statistics Policy (Required):
Please report SDs rather than SEMs.

Comments to the Author:

Your manuscript has been reviewed by two expert referees and a reviewing editor, and there was general agreement that the the main conclusions of the study are supported by the data that are presented. The study also has the potential to be impactful. However, both referees had a number of major concerns that would need to be addressed in a revised manuscript. Most of the concerns can be addressed with changes in the text. However, Reviewer 1 raises a number of points that would require additional experiments, as outlined in their major points 1-3. Of these, the editors believe that the most critical would be studies using different stimulation paradigms that address differences in results that were obtained here versus a prior study. We look forward to receiving a revised manuscript that addresses the concerns.

REFEREE COMMENTS

Referee #1:

Please see attached file.

Referee #2:

This manuscript by Koen & Maarten reports very interesting data about the role of mitochondria in calcium dynamics at the axon inner segment (AIS) in layer 5 pyramidal neurons. Using electron microscopy and virally delivered genetically encoded tools, the Authors show very clearly that at the AIS of pyramidal neurons, the site where action potentials (APs) initiate, cytoplasmic Ca²⁺-influx is very well buffered by proximally clustered mitochondria. This implies that the trains of APs generated at the AIS and recorded in situ by the patch-clamp technique are not significantly altered during high-frequency sustained neuronal activity. Overall, the conclusions are convincing and supported by excellent optical and electrophysiological recordings.

I do not have major criticisms but the text in several places needs to be more accurate and detailed so that a broad audience of readers may understand the logic of the experiments:

1) The block of the mitochondrial calcium uniporter (MCU) by Ru360 is shown not to alter the generation of the first evoked AP and the frequency of the train (Fig. 6 A-C). Of relevance in the figure is the recording of the single AP elicited with a brief current pulse (3 ms) (panel 6J). Here is very evident the presence of a "kink", which is typically generated at the AIS and that generates the early "bump" (the IS-spike) in the phase plan plot (PPP) of panel 6K. Convincingly, RU360 does not alter both the kink and the early part of PPP (the IS-spike). The Authors should underline more carefully this issue in the Results and the Discussion and comment also on why the IS-spike in the PPP is so large compared to the second AP in the PPP. Is there any particular Nav channel in layer 5 pyramidal neurons that is responsible for such behavior?

2) An important conclusion in the manuscript is that blocking mt-Ca²⁺ with RU360 does not alter the AP parameters of evoked high-frequency trains (100 Hz; Fig. 5E,H). Showing the details of the normalized AHP to the right of panel E is not sufficient. To be more convincing, the Authors should show the overlapping of APs in control and with RU360 as they appear and then after normalization as in panel 5E-right. It would be nice in this case to show also the time course of APs at higher time resolution so that the reader could also understand how the pacemaking is not altered by mt-Ca²⁺ block.

3) The authors do not mention the role of SK channels in layer 5 pyramidal neurons. Are these channels present in these neurons? SK channels are highly sensitive to cytoplasmic Ca²⁺ and may reveal more precisely any alteration of mitochondrial Ca²⁺ buffering at the AIS. Please clarify this issue!

END OF COMMENTS

This study presents a phenotypic examination of mitochondria in the axon initial segment (AIS) of layer 5 tufted pyramidal neurons in primary somatosensory cortex of adult mouse. Using a combination of electrophysiology and multiphoton confocal imaging, the authors present data showing that while mitochondria at the AIS are highly responsive to activity-dependent elevations in cytosolic Ca^{2+} by loading Ca into the mitochondrial matrix via the MCU complex, this response does not affect electrical properties of the AIS or neuronal firing. Localization and measurement of mitochondria is confirmed using light microscopy, and correlated to a preexisting data set of electron microscopy images from adult mouse visual cortex. MCU uptake was disrupted experimentally by acute (45min) intracellular dialysis of the inhibitor Ru360.

This study will have limited impact, in my opinion. Mitochondrial uptake of calcium ions has been well documented, and it would be surprising if there was no signal detected using the high affinity mt-GCAMP6f reporter. This study does not replicate a previously reported MCU-dependent increase in the slow afterhyperpolarization (sAHP), using a different protocol than that used previously. A lack of experimental detail further complicates interpretation of the findings.

A few additional experiments, and clarification of the results and approaches used, will improve this manuscript. Suggestions are listed as major and minor revision suggestions, below.

Major:

- 1) Statements that this study does not replicate prior work from Groten & McVicar, 2022; Kirchner 2024 is not wholly representative, as different stimulation paradigms were used. This study and this major result would be improved if the same stimulation paradigm (frequency, duration) were applied here as used in the prior work. This will resolve whether there are true cell-type differences between vasopressin magnocellular neurons (Kirchner), juvenile rat cortical pyramidal neurons (Groten and McVicar), and those studied here.
- 2) Since it has been shown that there is a second Ca^{2+} reservoir at the AIS of these cells (Discussion, bottom of p. 18), it would be worthwhile to acutely inhibit the saccular organelle in addition to the MCU, and see if there are compensatory effects between these two Ca^{2+} reservoirs. This experiment would provide clarity on mechanism occurring at this AIS that may differentiate it from neurons that have been studied.
- 3) Similarly, the experiment proposed in the first paragraph on p.20 of the Discussion could be feasibly accomplished with dialysis of Ru360. Up to 45 min dialysis was shown in figure 6, and genetically labeled mt-GFP were shown in Fig 2, so is feasible with the tools on hand.

- 4) Figure 6 is unclear. Panel B suggests that control (pipette internal dialysis versus Ru360 dialysis, but figure legend states traces were compared between break-in and after 45 min Ru360 dialysis. Text in results section for Figure 5 suggest that "Control" are infused (assumed for same time course) with pipette intracellular solution, while Ru360 are dialyzed with the same solution containing 20 μ M Ru360. If a different approach was used in Figure 6, this needs clarification and has important consequences for interpretation of the data.
- 5) Major concern regarding electrical access for dyes and especially Ru360. If pipettes are pulled at 6-7 M Ω , access resistance should be monitored, to ensure sufficient access is present for drug dialysis which will affect interpretation of results. Access resistance (average for recordings, and maximum allowable) needs to be reported in Materials and Methods. A lack of drug dialysis or high access resistance could partially explain the negative electrophysiology results found in figures 5 and 6.

Minor:

- 6) Typographical/grammar error on p.3, second paragraph, second sentence.
- 7) Mito sizes, especially for stationary mitos, are likely clusters of particles, not single events. This is particularly likely given the use of 2-p microscopy in these experiments (figure 3, figure 4), but also likely for standard confocal (Figure 2). Manuscript would be improved by referring to these objects as mitochondrial particles, or clusters, and not as individual mitochondria.
- 8) Fig 3: how was cellular contour identified in these experiments, if no secondary label (dye fill or other marker) was used? Please clarify.
- 9) Figure legend for figure 6, panels C and D are reversed, I think. Similarly, panels L-N are out of sync with the figure legend.
- 10) Mouse lines used are unclear. Both RRID:MMRRC_037128-UCD and MMRRC:032115 are listed in Materials and Methods.
- 11) Abbreviation ALS is not needed (p. 20), only used twice in the manuscript.
- 12) Sentence in the second paragraph of p. 20, "Using a nerve injury model..." is unclear, should be rewritten.

We thank the referees and editors for their constructive input to our manuscript. Here, we have provided a point-by-point response with the reviewers' text in black italic and our responses in blue. In the revised manuscript our changes are colored in red.

Reviewing Editor:

Data is presented as SEM, needs to be presented as SD

Thank you for the advice. We have revised the figures to report standard deviations.

This manuscript focuses on mitochondria regulation in calcium dynamics at the AIS at layer 5 pyramidal neurons. Both reviewers found the conclusions largely supported by the data reported in the manuscript. However, both reviewers have concerns about the manuscript. Both reviewers point out that significant text revisions are needed to better explain logic of experiments and expand the discussion section in the article. In addition, reviewer#1 pointed out to strengthen conclusions there is a need to determine if different stimulation paradigms between the studies could explain the difference in results between the studies.

Therefore, based on these positive comments and critiques address concerns in this manuscript an additional experiment that uses a similar stimulation paradigm as in previous studies to resolve what underpins the contrasting results is needed.

For the revision we have generated new experiments to examine the different protocols in the role of mitochondria to the AHP. In brief, we could confirm a role for mitochondrial Ca^{2+} uptake to the slow AHP.

Furthermore, a significant revision of the text will need to be made, and better discussion of the differences between this study and prior studies. Finally, the manuscript will need to comply with Journal policies for reporting of statistics as SD.

Based, in part, on the new findings we have revised the manuscript and improved the line of arguments in the discussion. There is no discrepancy between previous studies and ours and the main findings that mitochondria in the AIS powerful take up Ca^{2+} but do not contribute to local excitability for AP initiation remains. As indicated above, SD is shown in the figures.

Senior Editor:

Please report SDs rather than SEMs.

Thank you for the advice. We have revised the figures to report standard deviations.

Your manuscript has been reviewed by two expert referees and a reviewing editor, and there was general agreement that the the main conclusions of the study are supported by the data that are presented. The study also has the potential to be impactful. However, both referees had a number of major concerns that would need to be addressed in a revised manuscript. Most of the concerns can be addressed with changes in the text. However, Reviewer 1 raises a number of points that would require additional experiments, as outlined in their major points 1-3. Of these, the editors believe that the most critical would be studies using different stimulation paradigms that address differences in results that were obtained here versus a prior study. We look forward to receiving a revised manuscript that addresses the concerns.

In the revision we have integrated all comments and made new experimental data for major point 1 from referee #1. In brief, using the specific protocol used by Groten & McVicar we observed a significant contribution of the mitochondrial Ca^{2+} uptake in shaping the slow afterhyperpolarization. We conclude that

Referee #1:

This study presents a phenotypic examination of mitochondria in the axon initial segment (AIS) of layer 5 tufted pyramidal neurons in primary somatosensory cortex of adult mouse. Using a

combination of electrophysiology and multiphoton confocal imaging, the authors present data showing that while mitochondria at the AIS are highly responsive to activity-dependent elevations in cytosolic Ca^{2+} by loading Ca^{2+} into the mitochondrial matrix via the MCU complex, this response does not affect electrical properties of the AIS or neuronal firing. Localization and measurement of mitochondria is confirmed using light microscopy, and correlated to a preexisting data set of electron microscopy images from adult mouse visual cortex. MCU uptake was disrupted experimentally by acute (45min) intracellular dialysis of the inhibitor Ru360. This study will have limited impact, in my opinion. Mitochondrial uptake of calcium ions has been well documented, and it would be surprising if there was no signal detected using the high affinity mt-GCAMP6f reporter. This study does not replicate a previously reported MCU-dependent increase in the slow afterhyperpolarization (sAHP), using a different protocol than that used previously. A lack of experimental detail further complicates interpretation of the findings.

A few additional experiments, and clarification of the results and approaches used, will improve this manuscript. Suggestions are listed as major and minor revision suggestions, below.

We thank the reviewer for their suggestions for additional experiments and textual clarifications. For our revision of the manuscript, we have performed new electrophysiological experiments which we believe significantly improve the manuscript. Below we address each major and minor point raised by the reviewer.

Major:

1) Statements that this study does not replicate prior work from Groten & McVicar, 2022; Kirchner 2024 is not wholly representative, as different stimulation paradigms were used. This study and this major result would be improved if the same stimulation paradigm (frequency, duration) were applied here as used in the prior work. This will resolve whether there are true cell-type differences between vasopressin magnocellular neurons (Kirchner), juvenile rat cortical pyramidal neurons (Groten and McVicar), and those studied here.

We agree that the used stimulation protocol may have important implications for our results and conclusions and have therefore performed new experiments using the exact paradigms from Kirchner et al. 2024, and Groten & McVicar, 2022 (i.e. 30 APs at 20 Hz, and 200 APs at 20/50 Hz, respectively). We also increased the number of recordings for our paradigm of 100 APs at 100 Hz. The results are shown in revised Figure 5 and show that at higher frequencies (i.e. 50 or 100 Hz), Ru360 leads to a significantly longer sAHP duration (half-width), in line with Groten & McVicar (2022). Although Kirchner et al. (2024) found such differences also at lower frequencies, we could not replicate these results in L5 pyramidal neurons. These results are in support of the experiments by Groten & McVicar (2022), and suggest there are cell-type specific differences in the sAHP.

2) Since it has been shown that there is a second Ca^{2+} reservoir at the AIS of these cells (Discussion, bottom of p. 18), it would be worthwhile to acutely inhibit the saccular organelle in addition to the MCU, and see if there are compensatory effects between these two Ca^{2+} reservoirs. This experiment would provide clarity on mechanism occurring at this AIS that may differentiate it from neurons that have been studied.

The notion that Ca^{2+} release from the giant saccular organelle (GSO) could complement the role of mt- Ca^{2+} buffering at the AIS is indeed very interesting. However, such an experiment would ideally also be accompanied by additional ultrastructural and molecular characterization of the GSO at the AIS (similar to what we have done for mitochondria). Since our study focused on mitochondria we believe such experiments are beyond the scope of the current manuscript.

3) Similarly, the experiment proposed in the first paragraph on p.20 of the Discussion could be feasibly accomplished with dialysis of Ru360. Up to 45 min dialysis was shown in figure 6, and genetically labeled mt-GFP were shown in Fig 2, so is feasible with the tools on hand.

We have performed additional experiments to induce AIS plasticity in slices by exposure to NMDA based on the protocols used by Fréal et al. (Sci. Adv., 2023, doi: 10.1126/sciadv.adf3885). We combined this treatment with Ru360 (or control) to reveal any effect of mt-Ca²⁺ during AIS plasticity. However, the results showed a lack of AIS plasticity in L5 pyramidal neurons (see Supporting Figure 1 below). Although this does not preclude a role for mt-Ca²⁺ buffering in AIS plasticity in other cell types, given that all our other data was collected in L5 pyramidal neurons, we did not continue this research or include it in the current manuscript. The lack of activity-dependent AIS plasticity is in line with Jamann et al. using in vivo experiments showing that L5 pyramidal neuron AISs were stable while L2/3 pyramidal neuron AISs showed activity dependent length changes (Nat. Comm., 2021, doi: 10.1038/s41467-020-20232-x). In the revised version we discuss these points.

Supporting Figure 1 - No effect of NMDA alone or NMDA + Ru360 on L5 pyramidal neuron AIS length. Acutely prepared slices were exposed to 20 μM Ru360 (or control) for 45 minutes prior to and during subsequent 20 μM NMDA (or control) treatment. Control -Ru360, $28.25 \pm 3.80 \mu\text{m}$ ($n = 89$ AISs); NMDA -Ru360, $27.26 \pm 3.94 \mu\text{m}$ ($n = 54$ AISs); Control +Ru360, $28.35 \pm 3.26 \mu\text{m}$ ($n = 31$ AISs); NMDA +Ru360, $27.81 \pm 3.79 \mu\text{m}$ ($n = 35$ AISs).

4) Figure 6 is unclear. Panel B suggests that control (pipette internal dialysis versus Ru360 dialysis, but figure legend states traces were compared between break-in and after 45 min Ru360 dialysis. Text in results section for Figure 5 suggest that “Control” are infused (assumed for same time course) with pipette intracellular solution, while Ru360 are dialyzed with the same solution containing 20 μM Ru360. If a different approach was used in Figure 6, this needs clarification and has important consequences for interpretation of the data.

We thank the reviewer pointing out the lack of clarity in Figure 6. Except for Figure 6C-D, where we compare break-in vs. infused using a pair-wise statistical test, all comparisons were made between cells that were treated either with control or Ru360 (indeed the same time course), *i.e.* unpaired between control and Ru360. We hope to have made this clearer in the revised version of the figure 6 by plotting paired data with lines.

5) Major concern regarding electrical access for dyes and especially Ru360. If pipettes are pulled at 6-7 MOhm, access resistance should be monitored, to ensure sufficient access is present for drug dialysis which will affect interpretation of results. Access resistance (average for recordings, and maximum allowable) needs to be reported in Materials and Methods. A lack of drug dialysis or high access resistance could partially explain the negative electrophysiology results found in figures 5 and 6.

We agree that this is an important technical aspect and have now included specific information about access resistance in the Methods section. Note that the revised Figure 5 shows a significant impact of

Ru360 on the AHP, further indicating that poor infusion does not explain the lack Ru360 on action potential initiation as we describe in Figure 6.

Minor:

6) *Typographical/grammar error on p.3, second paragraph, second sentence.*

We were not entirely sure which error the reviewer was referring to but hope the revised sentence is clearer.

7) *Mito sizes, especially for stationary mitos, are likely clusters of particles, not single events. This is particularly likely given the use of 2-p microscopy in these experiments (figure 3, figure 4), but also likely for standard confocal (Figure 2). Manuscript would be improved by referring to these objects as mitochondrial particles, or clusters, and not as individual mitochondria.*

We agree with the reviewer that the resolution that 2P or confocal microscopy offers does not allow us to identify individual mitochondria with certainty, as also highlighted by the EM data which shows mitochondria clustering closely together. We have now explicitly highlighted this technical constraint in the main text.

8) *Fig 3: how was cellular contour identified in these experiemtns, if no secondary label (dye fillor other marker) was used? Please clarify.*

As can be seen in Figure 1A, mitochondria are prominently present in neuronal somata and the AIS. As such, a secondary label was not required to identify somata and the corresponding AIS. This was mentioned in the original main text but we have added a clarifying sentence in the legend of Figure 3. In addition, we also used brightfield light to confirm the location of (healthy appearing) somata and the corresponding AIS. We have added this to the Methods section.

9) *Figure legend for figure 6, panels C and D are reversed, I think. Similarly, panels L-N are out of syncwith the figure legend.*

We thank the reviewer for this feedback. We agree and have adjusted the legend.

10) *Mouse lines used are unclear. Both RRID:MMRRC_037128-UCD and MMRRC:032115 are listed in Materials and Methods.*

We have added these MMRRC line names in accordance with the acknowledgment instructions provided by MMRRC. From the website *"This congenic strain carries the same mutation as MMRR:032115 and was originally characterized on a STOCK background before backcrossing to C57BL/6J for at least 7 generations."*

11) *Abbreviation ALS is not needed (p. 20), only used twice in the manuscript.*

We have removed the abbreviation.

12) *Sentence in the second paragraph of p. 20, "Using a nerve injury model..." is unclear, should be rewritten.*

We agree that this sentence was unclear, we have adjusted it to improve its readability.

Referee #2:

This manuscript by Koen & Maarten reports very interesting data about the role of mitochondria in calcium dynamics at the axon inner segment (AIS) in layer 5 pyramidal neurons. Using electron microscopy and virally delivered genetically encoded tools, the Authors show very clearly that at the AIS of pyramidal neurons, the site where action potentials (APs) initiate, cytoplasmic Ca²⁺influx is

very well buffered by proximally clustered mitochondria. This implies that the trains of APs generated at the AIS and recorded *in situ* by the patch-clamp technique are not significantly altered during high-frequency sustained neuronal activity. Overall, the conclusions are convincing and supported by excellent optical and electrophysiological recordings.

I do not have major criticisms but the text in several places needs to be more accurate and detailed so that a broad audience of readers may understand the logic of the experiments:

We thank the reviewer for their positive evaluation and have revised the manuscript to improve clarity.

1) The block of the mitochondrial calcium uniporter (MCU) by Ru360 is shown not to alter the generation of the first evoked AP and the frequency of the train (Fig. 6 A-C). Of relevance in the figure is the recording of the single AP elicited with a brief current pulse (3 ms) (panel 6J). Here is very evident the presence of a "kink", which is typically generated at the AIS and that generates the early "bump" (the IS-spike) in the phase plan plot (PPP) of panel 6K. Convincingly, RU360 does not alter both the kink and the early part of PPP (the IS-spike). The Authors should underline more carefully this issue in the Results and the Discussion and comment also on why the IS-spike in the PPP is so large compared to the second AP in the PPP. Is there any particular Nav channel in layer 5 pyramidal neurons that is responsible for such behavior?

We agree with the reviewer that the lack of an effect of Ru360 on the AIS components of the AIS is a strong indicator for a lack of impact of mt-Ca²⁺ buffering at the AIS on AP initiation. In the revised manuscript we have described this with more clarity. In addition, we have adjusted the example trace of the single AP in Figure 6J to visualize the rising phase of the AP. The amplitude of the AIS spike we report here (~400 versus 600 V/s for the somatic component) is in line with previous experimental literature and models (e.g. Fig. 8 in Hanemaaijer et al. 2020).

2) An important conclusion in the manuscript is that blocking mt-Ca²⁺ with RU360 does not alter the AP parameters of evoked high-frequency trains (100 Hz; Fig. 5E,H). Showing the details of the normalized AHP to the right of panel E is not sufficient. To be more convincing, the Authors should show the overlapping of APs in control and with RU360 as they appear and then after normalization as in panel 5E-right. It would be nice in this case to show also the time course of APs at higher time resolution so that the reader could also understand how the pacemaking is not altered by mt-Ca²⁺ block.

We have adjusted Figure 5 according to the comments of the reviewer. The revised figure now clearly shows that there is no impact of Ru360 on the AP firing rate (pacemaking) and also displays a longer segment of the trace (in total 60 s duration compared to 5 s previously) to show the time course at which the membrane potential returns to baseline.

3) The authors do not mention the role of SK channels in layer 5 pyramidal neurons. Are these channels present in these neurons? SK channels are highly sensitive to cytoplasmic Ca²⁺ and may reveal more precisely any alteration of mitochondrial Ca²⁺ buffering at the AIS. Please clarify this issue!

Thank you for this question. SK channels in these specific neocortical pyramidal neuron subtypes have been reported in the dendrites by using apamin (Jones et al. eLife, 2017, <https://doi.org/10.7554/eLife.30333>) or at the soma and dendrites of pyramidal neurons using TRAM-34 (Roshchin et al., 2020, <https://doi.org/10.1038/s41598-020-71415-x>). Using Ca²⁺ uncaging the latter study showed also that there is no evidence for membrane expression of SK currents in the proximal axon.

Dear Dr Kole,

Re: JP-RP-2026-289290R1 "Robust activity-dependent mitochondrial calcium dynamics at the AIS is dispensable for action potential generation" by Koen Kole and Maarten H.P. Kole

Thank you for submitting your manuscript to The Journal of Physiology. It has been assessed by a Reviewing Editor and by 2 expert referees and we are pleased to tell you that it is acceptable for publication following satisfactory revision.

REVISION CHECKLIST:

We look forward to receiving your revised submission.

Yours sincerely,

Nathan Schoppa
Senior Editor
The Journal of Physiology

EDITOR COMMENTS

Reviewing Editor:

The authors have done a good job of responding to previous critiques. Please modify graphical abstract and tone down conclusion/inference on AIS mitochondria and intracellular cargo trafficking as pointed out by Reviewer #1.

Senior Editor:

Thank you for submitting your revised manuscript to Journal of Physiology. It has been reviewed by the two original referees along with a reviewing editor (RE), and we are pleased to let you know that the manuscript is now acceptable for publication pending your addressing a couple of points mentioned in the RE's report. There was some difference of opinion between the two referees, with reviewer 1 being more critical of some of the new experimental results presented in the revision. The editors appreciated the new results and believe that they add significantly to the study. Your next revision will be reviewed just by the RE.

REFEREE COMMENTS

Referee #1:

This study presents a phenotypic examination of mitochondria in the axon initial segment (AIS) of layer 5 tufted pyramidal neurons in primary somatosensory cortex of adult mouse. Using a combination of electrophysiology and multiphoton confocal imaging, the authors present data showing that while mitochondria at the AIS are highly responsive to activity-dependent elevations in cytosolic Ca²⁺ by loading Ca into the mitochondrial matrix via the MCU complex, this response does not affect electrical properties of the AIS or neuronal firing. Localization and measurement of mitochondria is confirmed using light microscopy, and correlated to a preexisting data set of electron microscopy images from adult mouse visual cortex. MCU uptake was disrupted experimentally by acute (45min) intracellular dialysis of the inhibitor Ru360.

Concerns regarding impact and reproducibility remain. Even using the same protocols and approached as earlier work (Groten and Macvicar 2022), results in this manuscript do not fully confirm that prior work, and show limited effect of Ru360. The impact of Ru360 block on AHP is very small and transient in the current work, in contrast to the paper in 2022. This raises concerns regarding efficacy of the major pharmacological reagent used (Ru360) which can be "dirty" even from commercial suppliers. The source of Ru360 or storage conditions were not provided in the manuscript. Concerns remain that weak Ru360 pharmacology could lead to false negative results on AP shape or membrane properties, and impact interpretation of results.

Impact on AHP is overstated in the graphical abstract.

Similarly, the inference that AIS mitochondria on e.g., intracellular cargo trafficking is made without positive results. Experiments showing the effect of selective impairment of mitochondrial function in the AIS on cargo trafficking are needed to support this claim.

The experiments were designed to test a specific hypothesis, whether mitochondrial Ca^{2+} buffering affects AP generation or shape in layer 5 cortical pyramidal cells. This hypothesis was not supported, and this negative results adds little to our understanding of neuronal physiology with respect to mitochondrial calcium buffering.

Referee #2:

The authors have fully answered to my criticisms. I do not have further comments.

END OF COMMENTS

EDITOR COMMENTS

Reviewing Editor:

The authors have done a good job of responding to previous critiques. Please modify graphical abstract and tone down conclusion/inference on AIS mitochondria and intracellular cargo trafficking as pointed out by Reviewer #1.

Thank you for sharing the comments from the referees and editors. In this revision we have toned down the intracellular cargo comments in the Summary, Abstract and Discussion. The changes in the manuscript are shown in red. The comment of Referee #1 about the graphical abstract seems unwarranted. Readers are not expecting it to be quantitative, since the data are in the manuscript. We found significant differences between the AHP and single AP changes and believe this is illustrated clearly in the graphical abstract.

Senior Editor:

Thank you for submitting your revised manuscript to Journal of Physiology. It has been reviewed by the two original referees along with a reviewing editor (RE), and we are pleased to let you know that the manuscript is now acceptable for publication pending your addressing a couple of points mentioned in the RE's report. There was some difference of opinion between the two referees, with reviewer 1 being more critical of some of the new experimental results presented in the revision. The editors appreciated the new results and believe that they add significantly to the study. Your next revision will be reviewed just by the RE.

Thank you for helping with the constructive evaluation of our manuscript, which was substantially improved with the new data.

REFEREE COMMENTS

Referee #1:

This study presents a phenotypic examination of mitochondria in the axon initial segment (AIS) of layer 5 tufted pyramidal neurons in primary somatosensory cortex of adult mouse. Using a combination of electrophysiology and multiphoton confocal imaging, the authors present data showing that while mitochondria at the AIS are highly responsive to activity-dependent elevations in cytosolic Ca²⁺ by loading Ca into the mitochondrial matrix via the MCU complex, this response does not affect electrical properties of the AIS or neuronal firing. Localization and measurement of mitochondria is confirmed using light microscopy, and correlated to a preexisting data set of electron microscopy images from adult mouse visual cortex. MCU uptake was disrupted experimentally by acute (45min) intracellular dialysis of the inhibitor Ru360.

Concerns regarding impact and reproducibility remain. Even using the same protocols and approached as earlier work (Groten and Macvicar 2022), results in this manuscript do not fully confirm that prior work, and show limited effect of Ru360. The impact of Ru360 block on AHP is very small and transient in the current work, in contrast to the paper in 2022. This raises concerns regarding efficacy of the major pharmacological reagent used (Ru360) which can be "dirty" even from commercial suppliers. The source of Ru360 or storage conditions were not provided in the manuscript. Concerns remain that weak

Ru360 pharmacology could lead to false negative results on AP shape or membrane properties, and impact interpretation of results.

We thank the referee for noting the company information was lacking. We now added this to the revised Methods and used –similar to Groten and MacVivar– the Ru360 from Sigma.

The comments about 'small effect' and reproducibility are puzzling. Our results and those from Groten and MacVivar show the same directionality: mt-Ca²⁺ block significantly increased the AHP duration. Differences in magnitude of change between the studies can have various reasons and we gladly provide some speculations. Groten and MacVivar recorded from a population of 'cortical pyramidal neurons' whereas we recorded from genetically identified L5 thick-tufted pyramidal neurons. Differences in size of the AHP between cell types have been reported earlier (Roshchin et al., 10.1038/s41598-020-71415-x and Guan et al. 10.1152/jn.00524.2014). The control AHP in Groten and MacVivar for the example recording in Fig. 6A is larger. Further differences could be that they recorded from rats whereas we used mice

Impact on AHP is overstated in the graphical abstract.

A graphical abstract is by definition not quantitative. Detailed information on the magnitude of change is presented in the Results. Since the difference was statistically significant it is logical to show the distinction between AHP and individual APs.

Similarly, the inference that AIS mitochondria on e.g., intracellular cargo trafficking is made without positive results. Experiments showing the effect of selective impairment of mitochondrial function in the AIS on cargo trafficking are needed to support this claim.

We agree, this was referring to the study of Tjiang and Zempel (2022, 10.1007/s00018-022-04150-3). We removed the cargo remark from the abstract and more clearly cited it in the Discussion.

The experiments were designed to test a specific hypothesis, whether mitochondrial Ca²⁺ buffering affects AP generation or shape in layer 5 cortical pyramidal cells. This hypothesis was not supported, and this negative results adds little to our understanding of neuronal physiology with respect to mitochondrial calcium buffering.

We tested our null hypothesis, and the evidence shows we could not reject it. Reporting 'negative' findings is valuable and the bias for positive results in the literature, both coming from editorial boards, referees and authors, is a known problem in research (see e.g. Heesen and Bright R. Soc. Open Sci., 2025, <https://doi.org/10.1098/rsos.240688>).

Referee #2:

The authors have fully answered to my criticisms. I do not have further comments.

We thank the referee for their helpful comments and time evaluating our manuscript.

Dear Professor Kole,

Re: JP-RP-2026-289290R2 "Robust activity-dependent mitochondrial calcium dynamics at the AIS is dispensable for action potential generation" by Koen Kole and Maarten H.P. Kole

We are pleased to tell you that your paper has been accepted for publication in The Journal of Physiology.

Yours sincerely,

Nathan Schoppa
Senior Editor
The Journal of Physiology

IMPORTANT POINTS TO NOTE FOLLOWING ACCEPTANCE OF YOUR PAPER:

- **IMPORTANT NOTICE ABOUT OPEN ACCESS:** To assist authors whose funding agencies mandate immediate public access to published research findings, The Journal of Physiology allows authors to pay an Open Access (OA) fee to have their papers made freely available immediately on publication.

- You can help your research get the attention it deserves! Check out Wiley's free Promotion Guide for best-practice recommendations for promoting your work at: www.wileyauthors.com/eoo/guide. You can learn more about Wiley Editing Services which offers professional video, design, and writing services to create shareable video abstracts, infographics, conference posters, lay summaries, and research news stories for your research at: www.wileyauthors.com/eoo/promotion.

- If you would like to receive our 'Research Roundup', a monthly newsletter highlighting the cutting-edge research published in The Physiological Society's family of journals (The Journal of Physiology, Experimental Physiology, Physiological Reports, The Journal of Nutritional Physiology and The Journal of Precision Medicine: Health and Disease), please click this link, fill in your name and email address and select 'Research Roundup':
<https://www.physoc.org/journals-and-media/membernews>

EDITOR COMMENTS

Reviewing Editor:

The authors have done an excellent job of addressing the previous critiques. There are no further concerns.

Senior Editor:

All concerns have been addressed. The manuscript is now acceptable for publication. Congratulations!